# The human cytomegalovirus protein UL147A downregulates the most prevalent MICA allele: MICA*008, to evade NK cell-mediated killing

**Einat Seidel**[1☯], **Liat Dassa**[1☯], **Corinna Schuler**[2], **Esther Oiknine-Djian**[3,4,5], **Dana G. Wolf**[3,4,5], **Vu Thuy Khanh Le-Trilling**[2☯*], **Ofer Mandelboim**[1☯*]

1 The Lautenberg Center for General and Tumor Immunology, The Faculty of Medicine, The Hebrew University Medical School, IMRIC, Jerusalem, Israel, 2 Institute for Virology of the University Hospital Essen, University Duisburg-Essen, Essen, Germany, 3 Clinical Virology Unit, Hadassah Hebrew University Medical Center, Jerusalem, Israel, 4 Department of Biochemistry, IMRIC, Jerusalem, Israel, 5 The Chanock Center for Virology, IMRIC, Jerusalem, Israel

☯ These authors contributed equally to this work.
* Khanh.Le@uk-essen.de (VTKL-T); oferm@ekmd.huji.ac.il (OM)

**Data Availability Statement:** All relevant data are within the manuscript and its Supporting Information files.

## Abstract

Natural killer (NK) cells are innate immune lymphocytes capable of killing target cells without prior sensitization. One pivotal activating NK receptor is NKG2D, which binds a family of eight ligands, including the major histocompatibility complex (MHC) class I-related chain A (MICA). Human cytomegalovirus (HCMV) is a ubiquitous betaherpesvirus causing morbidity and mortality in immunosuppressed patients and congenitally infected infants. HCMV encodes multiple antagonists of NK cell activation, including many mechanisms targeting MICA. However, only one of these mechanisms, the HCMV protein US9, counters the most prevalent MICA allele, MICA*008. Here, we discover that a hitherto uncharacterized HCMV protein, UL147A, specifically downregulates MICA*008. UL147A primarily induces MICA*008 maturation arrest, and additionally targets it to proteasomal degradation, acting additively with US9 during HCMV infection. Thus, UL147A hinders NKG2D-mediated elimination of HCMV-infected cells by NK cells. Mechanistic analyses disclose that the non-canonical GPI anchoring pathway of immature MICA*008 constitutes the determinant of UL147A specificity for this MICA allele. These findings advance our understanding of the complex and rapidly evolving HCMV immune evasion mechanisms, which may facilitate the development of antiviral drugs and vaccines.

## Author summary

Human cytomegalovirus (HCMV) is a common pathogen that usually causes asymptomatic infection in the immunocompetent population, but the immunosuppressed and fetuses infected *in utero* suffer mortality and disability due to HCMV disease. Current HCMV treatments are limited and no vaccine has been approved, despite significant

**Funding:** This study was supported by the ISF Israel-China grant (2554/18, https://www.isf.org.il/). Further support came from the GIF foundation (1412-414.13/2017, http://www.gif.org.il/), the ICRF professorship grant (https://www.icrfonline.org/), the Israeli Science Foundation (Moked, 442-18), a Ministry of Science Personal Medicine grant (3-14764) and the DKFZ-MOST grant (3-14931, https://www.dkfz.de/en/israel/index.html), all to O. M. E.S. was supported during her work by the Adams Fellowship Programme of the Israel Academy of Sciences and Humanities (https://adams.academy.ac.il/) and by the Foulkes Foundation (https://www.foulkes-foundation.org/). The funders had no role in study design, data collection and analysis, decision to publish, or preparation of the manuscript.

**Competing interests:** The authors have declared that no competing interests exist.

efforts. HCMV encodes many genes of unknown function, and virus-host interactions are only partially understood. Here, we discovered that a hitherto uncharacterized HCMV protein, UL147A, downregulates the expression of an activating immune ligand allele named MICA*008, thus hindering the elimination of HCMV-infected cells. Elucidating HCMV immune evasion mechanisms could aid in the development of novel HCMV treatments and vaccines. Furthermore, MICA*008 is a highly prevalent allele implicated in cancer immune evasion, autoimmunity and graft rejection. In this work we have shown that UL147A interferes with MICA*008's poorly understood, nonstandard maturation pathway, and acts additively with a functionally homologous HCMV protein, US9. Study of UL147A may enable manipulation of its expression as a therapeutic measure against HCMV.

## Introduction

Human cytomegalovirus (HCMV) is a betaherpesvirus with a large double stranded DNA genome of approximately 235 kilo base pair (kbp) [1]. HCMV encodes 170 canonical genes and recent work has additionally described noncanonical open reading frames, as well as several classes of small and large noncoding RNAs [2–7]. Of note, the functions of many of its genes remain unknown [8].

Infection with HCMV in healthy individuals is usually asymptomatic but results in lifetime persistence, due to HCMV's remarkable ability to evade host immune responses [9]. In immunosuppressed individuals, HCMV causes significant morbidity with a wide range of end-organ involvement, indirect complications such as increased graft rejection in transplant recipients and mortality. HCMV is also a common congenitally transmitted pathogen, which can cause sensorineural hearing loss and developmental delays in children born infected with HCMV [1,10]. The use of available drug treatments for HCMV is often limited by their significant toxicity [1], and though there are several promising candidates in development, no HCMV vaccine has been approved for use [11]. As a result, a better understanding of HCMV immune evasion mechanisms could aid in the development of novel treatments and vaccines that are urgently required [9].

Natural killer (NK) cells are lymphocytes belonging to the innate immune system, first characterized for their ability to kill cancer cells with no prior sensitization [12]. NK cells constitute a primary line of defence against virally infected cells, tumor cells, fungi and bacteria. They play a major role in controlling HCMV infection, as NK cell-deficient patients suffer lethal HCMV infections [13]. NK cell activation is governed by a balance of signals transduced by activating and inhibitory receptors [14]. When the balance tips in favour of activation, NK cells kill the target cells and secrete cytokines and chemokines that modulate the immune response [15,16].

A major activating receptor expressed on all NK cells is NKG2D, a c-type lectin that binds a family of eight stress induced ligands: MHC class I polypeptide-related sequences (MIC) A and B, and UL16 binding proteins (ULBP) 1–6 [17]. Healthy cells usually do not express these ligands, but they can be induced by different stresses including heat shock, DNA damage and viral infection [18].

MICA is the most polymorphic stress-induced ligand, with >150 known alleles encoding >90 different MICA proteins (http://hla.alleles.org/nomenclature/stats.html). Interestingly, the most prevalent MICA allele, MICA*008, contains a single nucleotide insertion in its transmembrane domain, giving rise to a short frameshifted sequence terminated by a premature stop codon. This truncated allele is first synthesized in a soluble form, but then becomes

tethered to the membrane by a glycosylphosphatidylinositol (GPI) anchor, through a slow and poorly characterized nonstandard maturation pathway [19]. In contrast, other MICA alleles are membrane-spanning proteins with a cytosolic tail. As a result, MICA*008 has unique biological properties, including different apical/basolateral sorting, preferential lipid raft localization and release by exosomes [19,20]. MICA*008 has been implicated in autoimmune disease, transplant rejection and cancer immune evasion [20–23].

HCMV targets the NKG2D stress-induced ligands by multiple and diverse immune evasion mechanisms, including by manipulation of the ubiquitin proteasome system [24–26]. The viral microRNA miR-UL112 inhibits MICB translation [27]. The viral protein UL16 sequesters MICB, ULBP1, ULBP2 and ULBP6 in the ER [28–32]. The viral MHC homologue UL142 retains MICA and ULBP3 in the *cis*-Golgi apparatus [33–35]. The viral proteins US18 and US20 jointly target MICA to lysosomal degradation, while US12, US13 and US20 jointly target ULBP2 and MICB [36–38]. Similarly, the viral protein UL148A induces the lysosomal degradation of MICA together with an unknown viral interaction partner [39].

Notably, MICA*008 is unaffected by these mechanisms, which gave rise to the hypothesis that it is highly prevalent due to the evolutionary advantage it confers [40]. However, we have previously found that the HCMV protein US9 specifically targets MICA*008 to proteasomal degradation, suggesting that HCMV has evolved counter-measures to this common allele. While studying US9-deficient HCMV, we discovered that MICA*008 is targeted by additional, as-yet unidentified HCMV factors, since even the US9-deficient virus could still downregulate MICA*008 to a certain degree [41].

UL147A is a 75 amino acid early-late HCMV protein that is unnecessary for viral replication and has no known function [42,43]. UL147A has a predicted N-terminal signal peptide, a predicted transmembrane domain at its C-terminus and no predicted glycosylation sites (https://www.uniprot.org/uniprot/F5H8R0) [44]. It is encoded in the UL*b'* region located at the right end of the unique long HCMV genomic segment. This 13–15 kbp region encompassing the UL133–UL150 genes is known to encode multiple immune evasion factors [33,34,39,40,45–51] and is present in clinical HCMV isolates but is frequently lost during serial passaging in fibroblast cell culture [7,43,52–55]. However, UL147A is highly conserved in clinical isolates with a maximum of 6–7% sequence variability [42]. Here, we show that UL147A specifically targets MICA*008 to proteasomal degradation, resulting in reduced NK-cell mediated killing of HCMV-infected cells.

## Results

### UL147A-deficient HCMV mutants are impaired in MICA*008 downregulation

Since the UL*b'* region of the HCMV genome contains genes that target full-length MICA alleles, we hypothesized that this region might also contain MICA*008-targeting genes. To search for such genes, VH3 human foreskin fibroblasts (HFF) that are MICA*008 homozygous [41] were uninfected, or infected with two variants of the HCMV strain AD169: AD169VarS (VarS), or a bacterial artificial chromosome (BAC)-cloned AD169VarL [56], named BAC2. AD169VarL and its BAC-cloned version BAC2 contain most of the UL*b'* genomic region except for a UL140–144 deletion, while VarS completely lacks UL*b'*. At 96 hours post-infection (hpi), MICA*008 surface expression was assayed by flow cytometry (Fig 1A). Interestingly, MICA*008 surface levels were upregulated following infection with VarS, but not with BAC2. This suggested that the UL*b'* region encodes at least one MICA*008-targeting gene. Furthermore, all these viruses contain US9 which itself downregulates MICA*008, suggesting the effect of the UL*b'* region was robust enough to be detected even with confounding from US9.

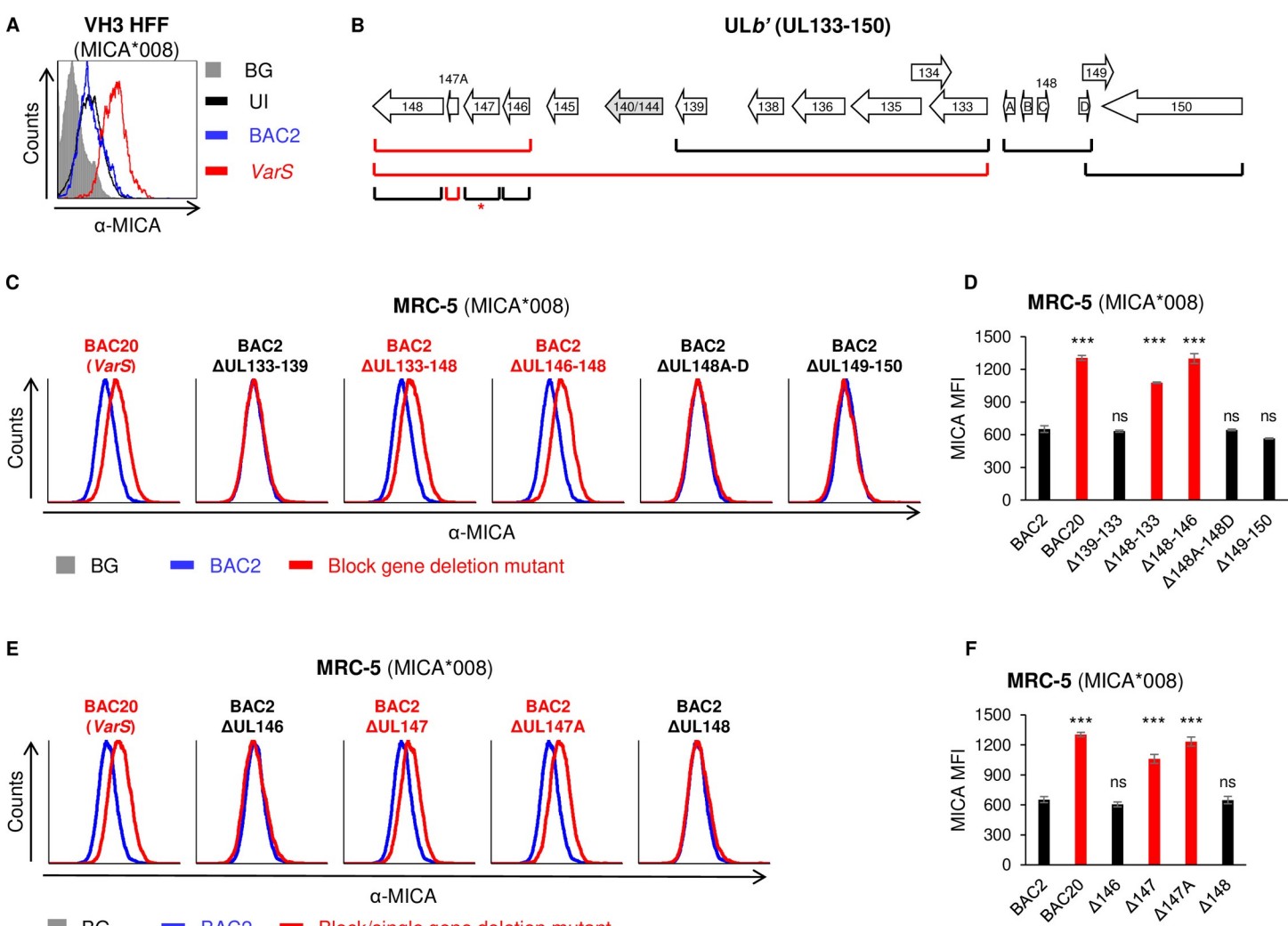

**Fig 1. UL147A-deficient HCMV mutants are impaired in MICA*008 downregulation.** (A) VH3 HFFs (MICA*008 homozygous) were either uninfected or infected with the indicated HCMV strains. Cells were harvested at 96 h post infection (hpi), and MICA surface expression was assayed by flow cytometry. Gray-filled histogram represents an isotype control staining of uninfected cells, all control stainings were similar to the one shown. (B) Diagram of the UL*b'* genomic region (UL133–150). Brackets indicate block or single gene deletions generated on the BAC2 background. Red brackets indicate deletion mutants impaired in MICA*008 downregulation. Red asterisk next to the BAC2ΔUL147 mutant indicates impaired MICA*008 downregulation caused by disruption of UL147A, as shown in S1 Fig. (C-F) MRC-5 HLFs (MICA*008 homozygous) were either uninfected or infected with the indicated HCMV strains. Cells were harvested at 72 h post infection (hpi), and MICA surface expression was assayed by flow cytometry. Shown are block deletion mutants (C) and their quantification (D) or single gene deletion mutants (E) and their quantification (F). Data show mean ±SEM for at least three independent repeats per mutant. A one-way ANOVA comparing all HCMV-infected cells was performed with a significant effect at the $p < 0.05$ level for all conditions [$F_{(10,73)} = 47.37$, $p = 6.7 \cdot 10^{-28}$]. MICA median fluorescence intensity (MFI) for BAC2 was compared to each mutant's MFI using a post-hoc Dunnett's test. Red font and bars highlight deletion mutants whose phenotype matches that of UL*b'*-deficient virus. *** $p < 0.001$, ns not significant. Full experimental data and statistics can be found in S1 Data.

To ascertain which UL*b'* gene(s) were responsible for the observed phenotype, we utilized an array of block and single-gene deletion mutants generated on the BAC2 background [39,49] (Fig 1B). Deletions in this region were previously shown to have no role in viral replication [49]. For this screen, we used MRC-5 human lung fibroblasts (HLFs) that are MICA*008 homozygous [57]. We assayed MICA surface expression by flow cytometry at 72 hpi. Initially, MRC-5 HLFs were infected with BAC-cloned VarS (termed BAC20), BAC2 or five different BAC2 block deletion mutants, to narrow our region of interest (Fig 1C, quantified in Fig 1D). Two of the block deletion mutants were impaired in MICA*008 downregulation, similarly to

BAC20: BAC2 ΔUL146-148, and the larger BAC2 ΔUL133-148 deletion that contains the UL146-148 deletion within it. In contrast, infection with BAC2 ΔUL133-139, BAC2 ΔUL148A-D and BAC2 ΔUL149-150 reduced MICA*008 surface expression to a similar extent as parental BAC2. We therefore focused on the UL146-148 genes for further analysis.

To identify the specific MICA*008-targeting gene(s), we generated BAC2 single deletion mutants of the four genes in this block: ΔUL146, ΔUL147, ΔUL147A and ΔUL148. We infected MRC-5 HLFs with BAC2, BAC20 and the single deletion mutants, and assayed MICA*008 surface expression at 72 hpi (Fig 1E, quantified in Fig 1F). Of the four single deletion mutants, both BAC2 ΔUL147A and BAC2 ΔUL147 were impaired in MICA*008 downregulation, similarly to BAC20.

This result was surprising for several reasons. First, there appeared to be no additive effect of each deletion (Fig 1E), arguing against two separate mechanisms. Second, UL147 is a viral CXC chemokine homolog [58] and therefore less likely to act as an MICA*008-targeting factor. Lastly, UL147 and UL147A are both expressed from the same transcript, with UL147A beginning 2nt downstream of the UL147 open reading frame (ORF). Due to these reasons, we decided to test the RNA expression levels of UL147 and UL147A in both BAC2 ΔUL147A and BAC2 ΔUL147. Northern blot analysis of RNA samples from uninfected, BAC2, ΔUL147 or ΔUL147A -infected MRC-5 cells, showed no detectable expression of UL147A in the ΔUL147 deletion mutant (S1A Fig). Conversely, the UL147 transcript was affected by the deletion of UL147A and appeared to be expressed at lower levels and altered transcript sizes. These results demonstrate that UL147A expression is disrupted both in ΔUL147 and in ΔUL147A.

To directly assess UL147's effect on MICA*008, we next transduced UL147 into RKO cells which overexpress MICA*008-HA. Indeed, MICA*008 surface expression levels remained constant during UL147 expression (S1B Fig). We therefore concluded that UL147A was the ULb' gene required for MICA*008 downregulation.

## Total MICA*008 protein quantity is reduced by UL147A overexpression

We next wanted to determine if UL147A is sufficient for MICA*008 downregulation. We cloned UL147A fused to an N-terminal FLAG tag that was inserted after its endogenous signal peptide. We transduced an empty vector (EV) control or the UL147A-FLAG construct into 293T cells (MICA*008 homozygous) or HCT116 cells (MICA*001/*009:02 full-length alleles) [59,60]. We then measured MICA surface expression by flow cytometry (Fig 2A). MICA*008 surface expression was significantly downregulated in UL147A-overexpressing 293T cells compared to EV controls, but there was no difference in MICA levels in HCT116 cells. This lack of effect on full-length MICA alleles matches our previous results, since we showed that the ΔUL146-148 block deletion mutant was not impaired in full-length MICA downregulation [39].

Next, we lysed the transfected cells and performed a western blot to visualize UL147A (Fig 2B). In both cell types, UL147A migrated as a single band of 12–13 kDa. We then assessed UL147A's effect on total MICA quantity. MICA is a highly glycosylated protein that migrates as a 'smear' of differentially glycosylated forms in western blots [41]. Importantly, whole-cell MICA*008 quantity was markedly reduced in 293T-UL147A cells, with no effect in HCT116-UL147A cells (Fig 2C). This suggested that the UL147A-mediated reduction in surface MICA*008 was not due to intracellular sequestration, but rather due to a reduction in total protein quantity.

We confirmed these results in RKO cells transduced with an EV, MICA*004-HA or MICA*008-HA. RKO cells endogenously express very low levels of intracellular MICA*007:01 and hence are a useful model for comparing exogenously expressed MICA alleles [41]. The various RKO cells were transduced with an EV control or with UL147A-FLAG, and MICA surface expression was measured by FACS staining (Fig 2D). Here too, only MICA*008 was

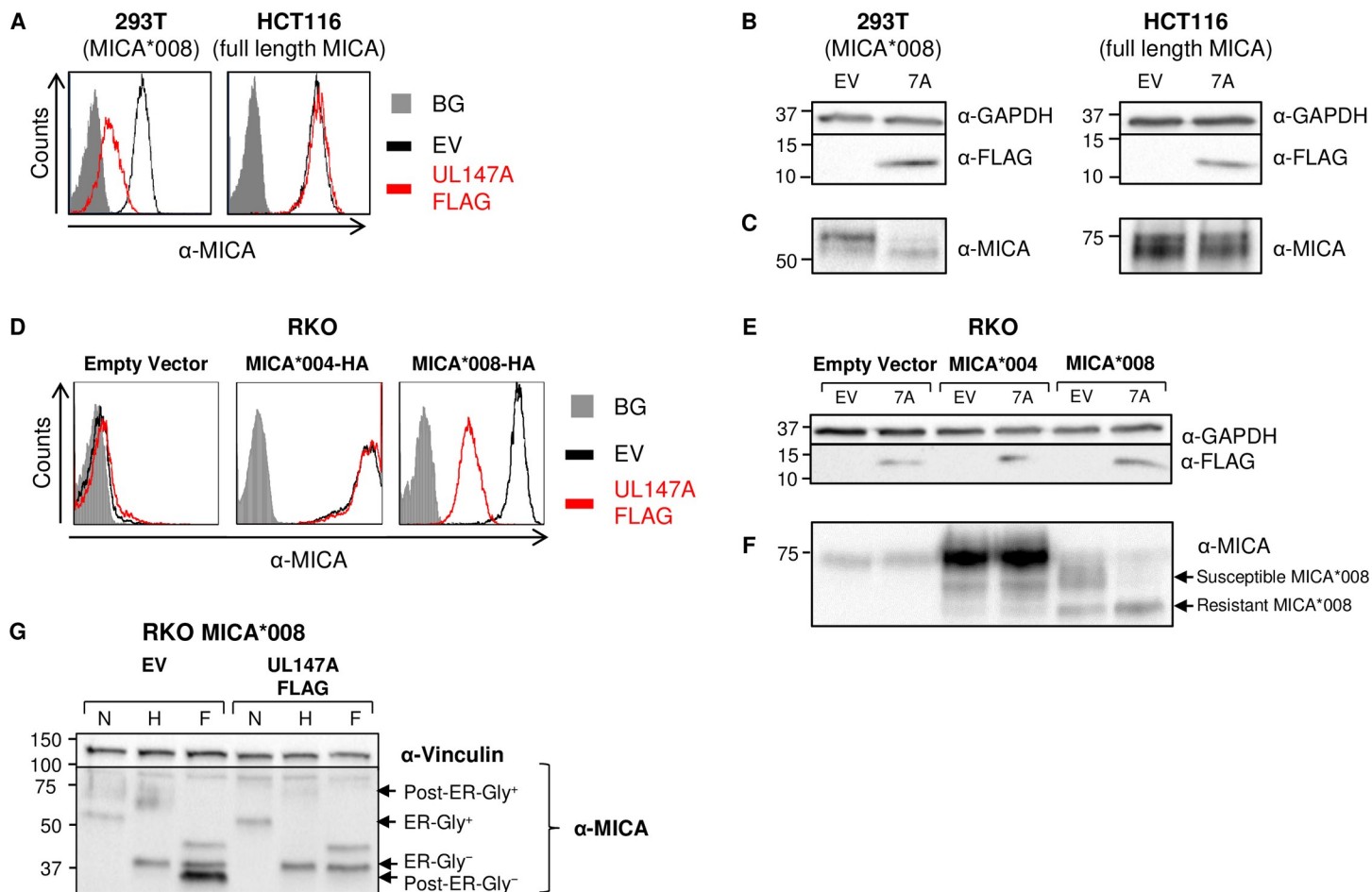

**Fig 2. Total MICA\*008 protein quantity is reduced by UL147A overexpression.** (A) 293T (MICA\*008 homozygous) and HCT116 (MICA\*001/\*009:02 full length alleles) were transduced with an EV (black histograms) or with UL147A-FLAG (red histograms) and MICA surface expression was assayed by flow cytometry. Gray-filled histograms represent secondary antibody staining of EV cells, all control stainings were similar to the one shown. Representative of three independent experiments. (B-C) The cells shown in (A) were lysed and a western blot was performed using anti-MICA antibody for detection of MICA, anti-FLAG tag antibody for detection of UL147A, and anti-GAPDH antibody as a loading control. The lysates were split in two and run on two gels (shown separately in B, C) to resolve proteins of different sizes. Representative of three independent experiments. (D) RKO cells were transduced with an EV, with MICA\*004-HA or with MICA\*008-HA, and then co-transduced with an EV or with UL147A-FLAG. MICA surface expression was assayed by flow cytometry. Gray-filled histograms represent secondary antibody staining of EV cells, all control stainings were similar to the one shown. Representative of three independent experiments. (E-F) The cells shown in (D) were lysed and a western blot was performed using anti-MICA antibody for detection of MICA, anti-FLAG tag antibody for detection of UL147A, and anti-GAPDH antibody as a loading control. The lysates were split in two and run on two gels (shown separately in E, F) to resolve proteins of different sizes. Arrows indicate UL147A-susceptible and UL147A-resistant forms of MICA\*008. Representative of three independent experiments. (G) RKO MICA\*008-HA cells co-transduced with EV or UL147A-FLAG were lysed. Lysates were left untreated or digested with endoH or with PNGaseF (marked N, H and F, respectively), and then blotted using anti-MICA and anti-vinculin as a loading control. Arrows indicate ER-resident (endoH sensitive) and post-ER (endoH resistant) MICA\*008 forms, with and without glycosylations (Gly$^{+/-}$). Representative of two independent experiments.

downregulated by UL147A while MICA\*004 remained unchanged. We also stained the RKO MICA\*008 cells for other NKG2D ligands and for MHC class I (S2 Fig) and found that they were unaffected by UL147A, suggesting it is MICA\*008-specific. Notably, a similar lack of effect of the UL*b'* region on other NKG2D ligands was previously observed during HCMV infection [39].

Next, we lysed the transfected RKO cells and performed a western blot analysis of UL147A expression (Fig 2E) and of total MICA quantity (Fig 2F). As with endogenous MICA, there was a substantial overall decrease in MICA\*008 quantity and no effect was observed on MICA\*004. However, not all MICA\*008 forms were equally affected by UL147A (Fig 2F,

forms indicated by arrows). A ~70 kDa band vanished, while a ~60 kDa band remained unchanged. A similar effect can be seen in the UL147A-expressing 293T cells (Fig 2C) but it is harder to appreciate since different MICA*008 forms migrate closely to each other in 293T cells. The sparing of certain MICA*008 protein forms indicated that UL147A affects MICA*008 at some point following its translation.

In RKO cells, different MICA*008 maturation stages can be distinguished relatively easily, since N-linked glycosylations first added at the ER are modified and expanded during passage through the Golgi apparatus, and the resulting increase in glycoprotein size is prominent compared to other cell types [41]. We therefore speculated that the differently sized UL147A-susceptible and UL147A-resistant MICA*008 forms correspond to different stages of MICA*008 maturation.

To directly identify the UL147A-resistant MICA*008 form, we performed an endoglycosidase H (endoH) sensitivity assay. EndoH only removes unmodified N-linked glycosylations prior to glycoprotein passage through the Golgi apparatus, while peptide N-glycosidase F (PNGaseF) removes all N-linked glycosylations. We digested lysates obtained from RKO MICA*008-HA cells expressing an EV or UL147A-FLAG (Fig 2G). In EV-expressing cells, most of MICA*008 was in the highly glycosylated ~70 kDa form, which was endoH-resistant. Only the minor, ~60 kDa band was endoH sensitive. Notably, following deglycosylation, the endoH-resistant form migrated more rapidly at ~34 kDa compared to the endoH-sensitive form which migrated at ~37 kDa. This apparent size difference is due to the presence of the GPI anchor in the mature, endoH-resistant form of MICA*008, which increases the negative charge of the glycoprotein [19].

In contrast, in UL147A-FLAG expressing cells, only the ~60 kDa endoH-sensitive form remained, indicating ER-resident MICA*008 is the UL147A-resistant form. UL147A's relative sparing of the ER-resident form could indicate that MICA*008 is being degraded just before or just after it exits the ER, or alternately, that MICA*008 is diverted from the secretory pathway and therefore fails to pass through the Golgi apparatus.

## UL147A is an ER-resident protein which reduces surface MICA*008 but spares ER-resident MICA*008

To directly visualize UL147A's effect on MICA*008 and determine the cellular localization of both proteins, we utilized immunofluorescence. We fixed and permeabilized RKO MICA*008-HA cells expressing either an EV control or UL147A-FLAG and stained them for the ER marker protein disulfide isomerase (PDI), for MICA and for FLAG tag (Fig 3A). Nuclei were counterstained with DAPI. Interestingly, UL147A co-localized extensively with PDI, indicating it is ER-resident. As for MICA*008, in EV-expressing cells, it resided mostly at the cell surface with a fraction co-localizing with the ER marker PDI. However, in UL147A-expressing cells, surface MICA*008 all vanished, and the remaining intracellular MICA*008 overlapped PDI, indicating UL147A-resistant MICA*008 was ER-resident. These results strengthen our findings from the endoH sensitivity assay (Fig 2G) since they demonstrate that mature MICA*008 is UL147A-susceptible. Importantly, these results also rule out the possibility of altered MICA*008 subcellular localization, suggesting it is being degraded at some point along its maturation pathway.

## UL147A targets ER-resident MICA*008 to proteasomal degradation prior to the GPI-anchoring step

Our results suggested that UL147A is ER-resident and that it induces MICA*008 degradation. To test whether UL147A targets MICA*008 to lysosomal or to proteasomal degradation, we performed a cycloheximide (CHX) chase assay in the presence of lysosomal and proteasomal

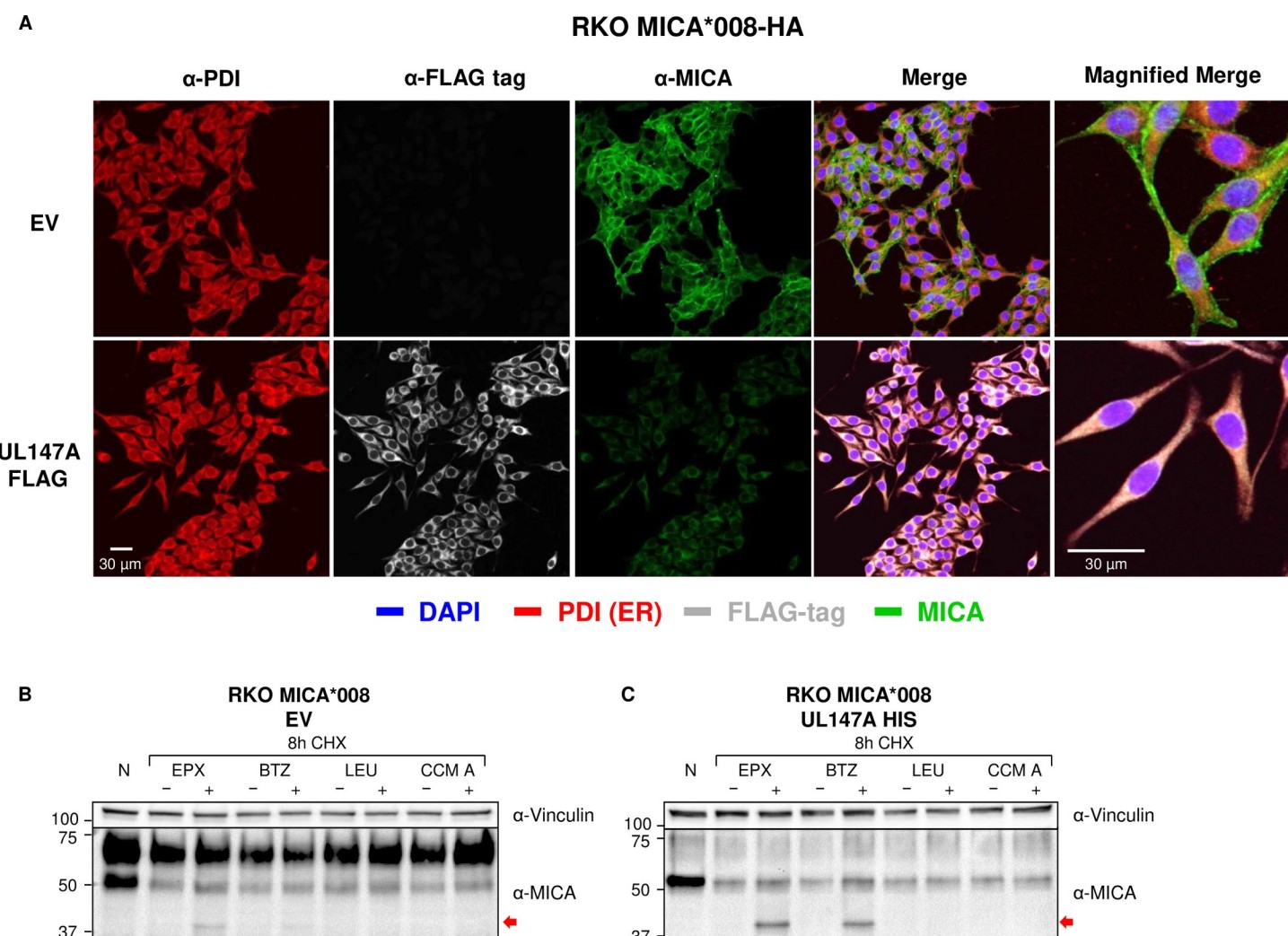

**Fig 3. UL147A is an ER-resident protein which reduces surface MICA\*008 but spares ER-resident MICA\*008.** (A) RKO MICA\*008-HA cells transduced with an EV or with UL147A-FLAG were grown on glass slides, fixed and stained with an anti-protein disulfide isomerase (PDI) antibody (ER marker; red), an anti-FLAG tag antibody (gray) and an anti-MICA antibody (green). Nuclei were stained with DAPI (blue). Images were captured by confocal microscopy. Representative of two independent experiments. (B-C) RKO-MICA\*008-HA cells expressing an EV (B) or N-terminally tagged UL147A (C) were left untreated (N), or incubated for 8 hours with the translation inhibitor cycloheximide (CHX, 50 µg/ml), in combination with one of two lysosomal inhibitors: leupeptin (LEU, 100 µg/ml) and concanamycin A (CCM A, 20 nM), or with one of two proteasomal inhibitors: epoxomicin (EPX, 8 µM) and bortezomib (BTZ, 8 µM). Each inhibitor was matched with an appropriate mock-treatment (DMSO or DDW). Following treatment, cells were lysed and blotted with anti-MICA. Anti-vinculin served as loading control. Representative of two independent experiments.

inhibitors. CHX globally inhibits protein translation, facilitating the study of protein degradation rates. RKO MICA\*008-HA cells expressing either an EV control (Fig 3B) or UL147A (Fig 3C) were left untreated or treated for 8 hours with CHX, in combination with the proteasomal inhibitors epoxomicin (EPX) or bortezomib (BTZ), or the lysosomal inhibitors leupeptin (LEU) or concanamycin A (CCMA). For each inhibitor, the appropriate vehicle-only mock treatment was matched.

In UL147A-expressing cells (Fig 3C), even after 8 hours of CHX treatment, ER-resident MICA\*008 remained detectable, demonstrating slow degradation kinetics. Importantly, treatment with proteasomal inhibitors resulted in the significant accumulation of a ~37 kDa form of MICA\*008, which represents a deglycosylated cytosolic degradation intermediate which is not GPI anchored, based on its size [41]. In contrast, lysosomal inhibitors did not affect

MICA*008 levels. The same degradation intermediate was faintly visible in EV-expressing controls treated with EPX or BTZ (Fig 3B), suggesting that low levels of MICA*008 were directed to proteasomal degradation even in UL147A's absence. We therefore concluded that UL147A directs MICA*008 to degradation in the proteasome after a prolonged lag in the ER, but prior to the GPI-anchoring step and ER exit.

## UL147A and US9 are functionally redundant in an overexpression system

Since the UL147A mechanism of action was extremely similar to that described for US9, which also targets MICA*008 to degradation prior to GPI anchoring [41], we decided to test the effect of co-expressing US9 and UL147A together. RKO MICA*008-HA cells were transduced with an EV control, US9-HIS (described in [41]), UL147A-FLAG, or co-transduced with both UL147A and US9. MICA*008 surface levels were then measured by flow cytometry (S3A Fig, quantified in S3B Fig). There was no significant difference in MICA*008 levels between cells expressing US9 alone, UL147A alone, or the two proteins together. We concluded that US9 and UL147A were redundant in an overexpression system, strengthening the conclusion that they are functionally homologous.

## Specific MICA*008 features are required for UL147A-mediated downregulation

We next wondered which MICA*008 features mediate UL147A's specificity to this allele. MICA*008 has two features distinguishing it from full-length MICA alleles: a frameshifted sequence of 15 amino acids, and a premature stop codon that terminates this sequence. We previously generated RKO cells transduced with MICA*004 mutants modified with different MICA*008 sequence features [41] (Fig 4A). In MICA*004-G-ins-HA, the G-nucleotide insertion of MICA*008 was introduced into the TM domain of MICA*004, and therefore this construct contains the 15 frameshifted amino acids and the premature stop codon. MICA*004-stop-HA only contains the premature stop codon at the same position as MICA*004-G-ins-HA. MICA*004-Dmut-HA is a double-mutated MICA*004. It includes the G-nucleotide insertion, but this insertion is corrected by a compensatory insertion after the 15 amino acid sequence, to restore the reading frame for the full remaining length of the MICA*004 allele. As a result, this allele has the frameshifted 15 amino acid sequence but not the premature stop codon. We previously found that only MICA*004-G-ins becomes GPI-anchored like MICA*008 [41]. To address the role of the GPI anchor, we previously generated a MICA*008 mutant named MICA*008-ULBP3TM-HA, in which we swapped the transmembrane domain with that of ULBP3, another NKG2D ligand that contains a canonical GPI-anchoring signal. This mutant is also GPI-anchored, but via the rapid, canonical pathway [41].

RKO cells were transduced with an EV control or with the various MICA constructs, and then co-transduced with an EV control or with UL147A-FLAG. Construct surface expression was measured by flow cytometry (Fig 4B). Intriguingly, of all the MICA mutants, UL147A downregulated only MICA*004-G-ins. This suggests that both the frameshifted sequence and the premature stop codon are required for UL147A-mediated effect. Both features together are also required for MICA*008 GPI anchoring. However, canonical GPI anchoring was not sufficient for UL147A recognition, since MICA*008-ULBP3TM levels were unaffected by UL147A. Taken together with the finding that MICA*008 degradation occurs before it becomes GPI-anchored, these results suggest that UL147A-mediated downregulation depends on MICA*008's non-canonical maturation pathway but not on the presence of the GPI anchor itself [41].

Having shown that the frameshifted sequence and premature stop codon were required features for UL147A recognition of MICA, we wanted to test whether they are also sufficient for

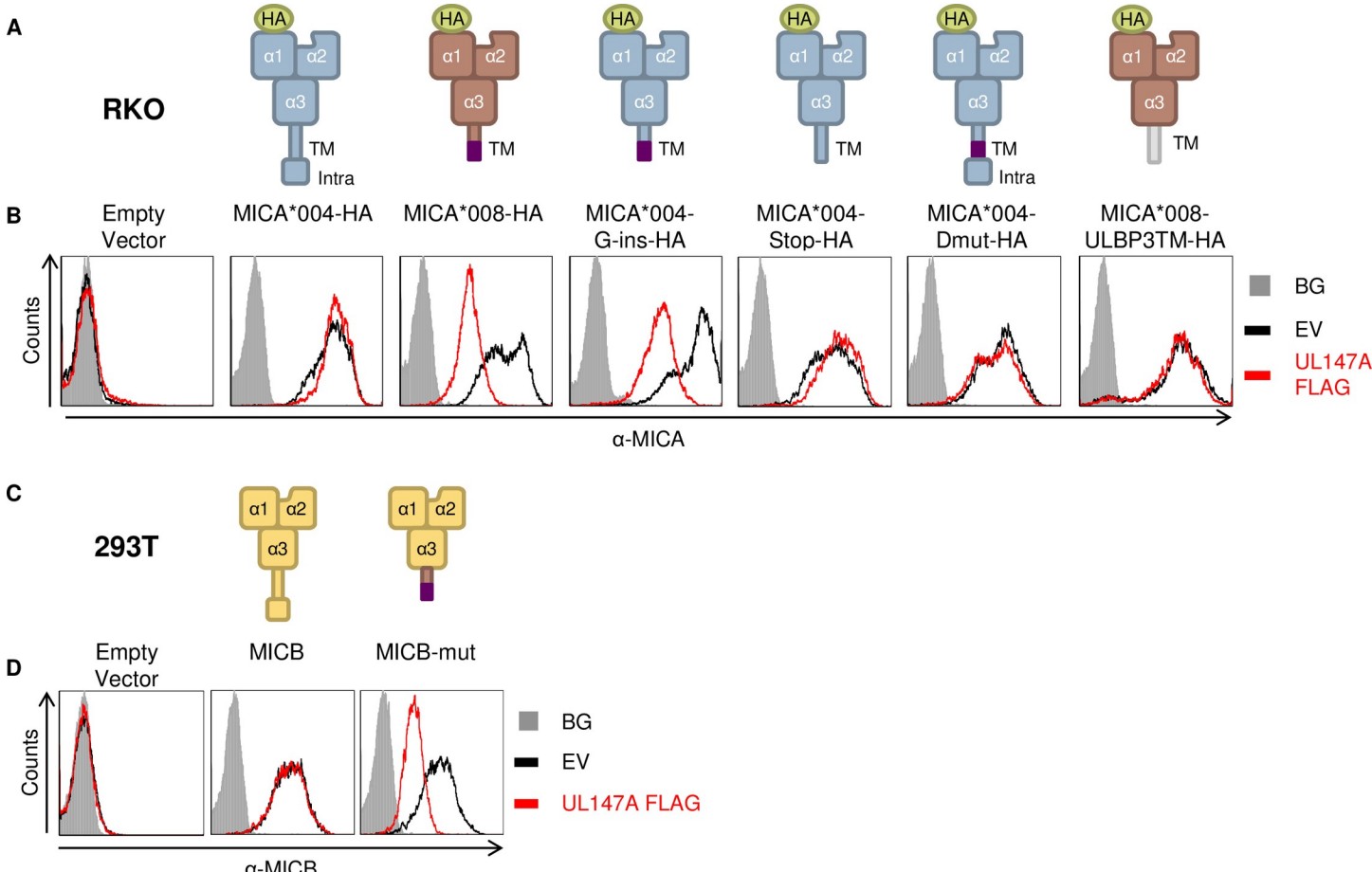

**Fig 4. Specific MICA\*008 features are required for UL147A-mediated downregulation.** (A) Schematic representation of the MICA mutants and chimeric proteins used to identify which feature of MICA*008 is recognized by UL147A. Annotated are the N-terminal HA tag, α1–3 domains, the transmembrane (TM) domain and the intracellular tail (intra). The frameshifted MICA*008 sequence is shown in purple. (B) FACS staining of MICA expression in RKO cells transduced with the MICA proteins described in (A) and co-transduced with an EV (black histogram) or with UL147A-FLAG (red histogram). Gray-filled histograms represent secondary antibody staining of EV cells, all control stainings were similar to the one shown. Representative of three independent experiments. (C) Schematic representation of MICB and a mutated MICB with MICA*008's TM domain. (D) Anti-MICB FACS staining of 293T cells transduced with an EV (left histogram); with WT MICB (middle histogram); or with MICB-mut (right) and co-transduced with an EV (black histogram) or with UL147A-FLAG (red histogram). Gray-filled histograms represent secondary antibody staining of EV cells, all control stainings were similar to the one shown. Representative of three independent experiments.

UL147A recognition in the context of a different protein: MICB, which is not targeted by UL147A. 293T cells, which lack endogenous MICB surface expression, were transduced with an EV control, WT MICB, or with MICB-mut where the endogenous MICB TM domain was swapped with the MICA*008 TM domain (Fig 4C) [41]. We then co-transduced the 293T cells with an EV control or with UL147A-FLAG, and assessed MICB surface expression using flow cytometry. Native MICB was unaffected by UL147A, but the mutant bearing MICA*008's TM domain was substantially downregulated by it (Fig 4D). This shows that MICA*008's TM domain is sufficient for conferring UL147A susceptibility.

## UL147A and US9 act additively during HCMV infection to induce intracellular MICA*008 retention

After characterizing UL147A's mechanism of action in an overexpression system, we wanted to assess its interactions with US9 and functional significance during HCMV infection. In

addition to the UL147A and US9 single deletion mutants on the BAC2 background, we also wanted to study a deletion mutant lacking both proteins. We therefore generated a US9 deletion mutant on the BAC20 background that lacks the entire ULb' region and therefore lacks UL147A, as we have shown that UL147A is the only MICA*008-targeting gene in this region (Figs 1E and S1). MRC-5 HLFs were uninfected or infected with BAC2, BAC2 ΔUL147A, BAC2 ΔUS9, BAC20 and BAC20 ΔUS9. A time course assay was performed to track MICA*008 surface levels at 24, 48 and 72 hpi by flow cytometry (Figs 5A and S4A, quantified in Figs 5B and S4B). Differences in MICA*008 levels were not significant at the 24 and 48 hpi time points (S4A and S4B Fig). At 72 hpi, cells infected with BAC2 ΔUS9, BAC2 ΔUL147A or BAC20 showed a three-to-four-fold increase in MICA*008 levels compared to BAC2-infected cells, with no significant differences between BAC2 ΔUL147A and the others (Fig 5A and 5B). Significantly, BAC20 ΔUS9-infected cells increased MICA*008 levels more than tenfold compared to BAC2-infected cells. In contrast, when we infected FLS1 HFFs (endogenous full-length MICA), there were no discernible differences between BAC2 and BAC2 ΔUL147A infected cells in MICA surface expression or in total MICA levels (S4C and S4D Fig), confirming UL147A's allele specificity also during infection. These results show that UL147A has a similar effect magnitude and timing to that of US9 on MICA*008 surface expression, and that deletion of the two has an additive effect during infection.

We previously reported that the bulk of MICA*008 was retained in the ER during HCMV infection, and that US9 did not lead to appreciable differences in total MICA*008 quantity in HCMV-infected cells. We attributed this effect to additional viral MICA*008-targeting mechanism(s) [41]. We therefore wondered whether UL147A would induce MICA*008 degradation during HCMV infection.

We infected MRC-5 HLFs as before, and lysed them at 72 hpi. Lysates were then analyzed by western blot (Fig 5C). In accordance with our previous results, in UI cells low levels of mostly post-ER MICA*008 (~60 kDa) were detected. A faster-migrating (~50 kDa), ER-resident form of MICA*008 accumulated in cells infected with BAC2, BAC2 ΔUL147A, BAC2 ΔUS9, and BAC20 (forms annotated in Fig 5C). Importantly, in cells infected with the BAC20 ΔUS9 mutant that lacks both US9 and UL147A, a portion of MICA*008 was redistributed as slowly-migrating forms (~70 kDa), implying that MICA*008 was exiting the ER. These results suggest that US9 and UL147A act primarily as ER-retention factors during HCMV infection.

We repeated these experiments in FLS3*008 HFFs, which lack endogenous MICA and overexpress MICA*008-HA [41]. Cells were uninfected or infected with the variuos HCMV strains and stained for MICA surface expression at 72 hpi (Fig 5D, quantified in Fig 5E). As in MRC-5 cells, MICA levels increased by two-to-five-fold in cells infected with BAC2 ΔUS9, BAC2 ΔUL147A or BAC20 compared to BAC2, though the effect for ΔUS9 was not statistically significant in this assay, possibly due to insufficient statistical power and/or smaller US9 effect size. Nonetheless, the differences between ΔUL147A and ΔUS9 or ΔUL147A and BAC20 were not significant. MICA levels increased by more than 15-fold in cells infected with double mutant BAC20 ΔUS9 compared to BAC2-infected cells, confirming the additive roles of US9 and UL147A during HCMV infection. Similar results were also obtained in western blot analysis (Fig 5F), demonstrating a redistribution of MICA*008 to a 'smear' of high molecular weight forms upon BAC20 ΔUS9 infection.

## UL147A and US9 arrest MICA*008 maturation and induce its degradation during HCMV infection

We next wanted to address MICA*008 maturation and degradation dynamics during HCMV infection. We therefore conducted a CHX chase assay. MRC-5 cells were uninfected or

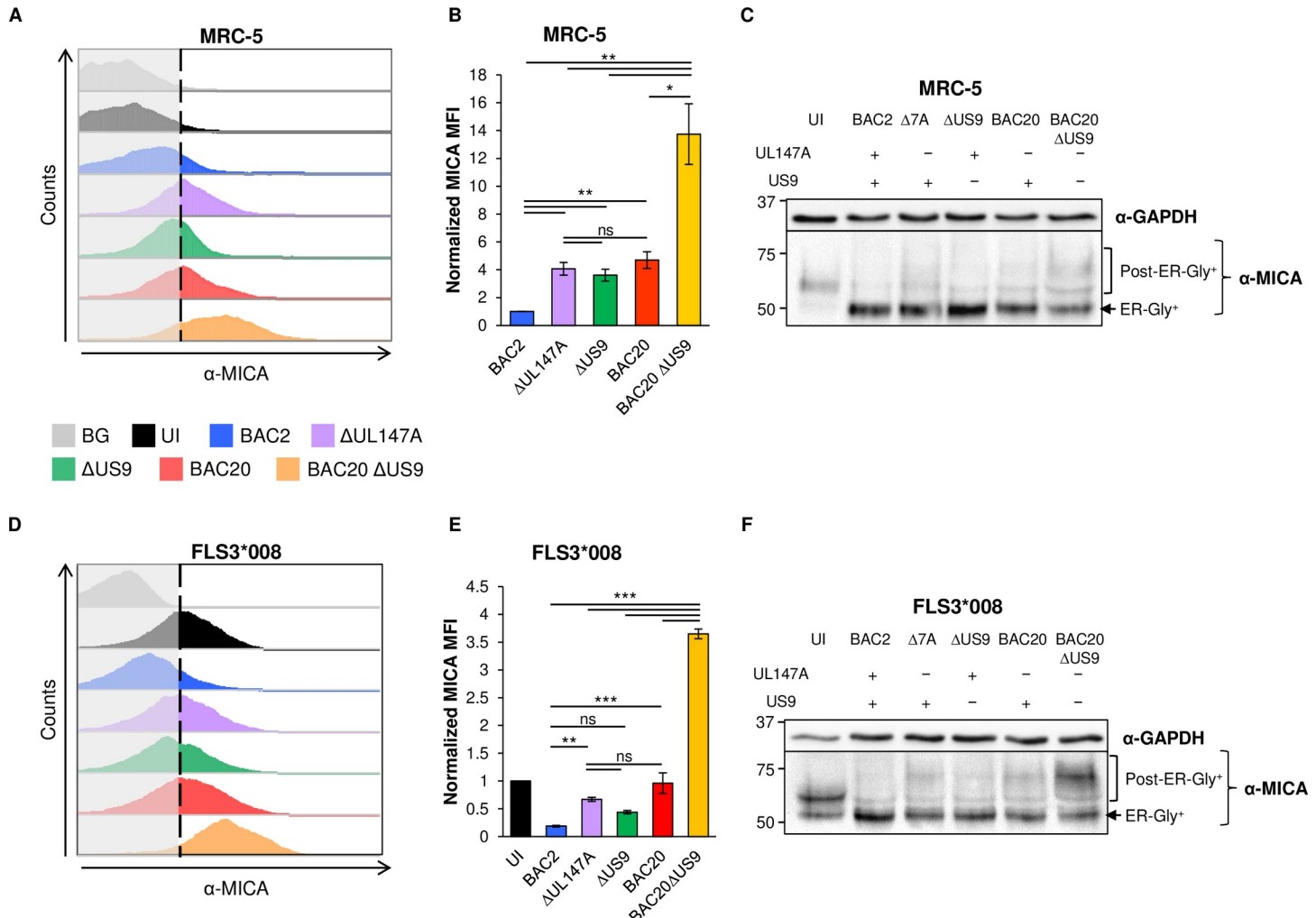

**Fig 5. UL147A and US9 act additively to reduce MICA\*008 surface expression by intracellular sequestration.** (A-C) MRC-5 HLFs (MICA\*008 homozygous) and (D-F) FLS3\*008 (overexpressing MICA\*008) were either uninfected (UI) or infected with the indicated HCMV strains. Cells were harvested 72 hours post infection (hpi). (A, D) MICA surface expression was assayed by flow cytometry. Gray-filled histograms represent a secondary antibody staining of UI cells, all control stainings were similar to the ones shown. Representative of seven independent experiments for MRC-5 and three independent experiments for FLS3\*008. (B, E) MICA median fluorescent intensity (MFI) values were quantified. Data show mean ±SEM. (B) Values were normalized to BAC2-infected cells. A repeated measures ANOVA with a Greenhouse-Geisser correction showed that MICA values differed significantly between infected cells [F (1.099,6.595) = 30.99, p = 0.0009]. (E) Values were normalized to uninfected cells and a one-way ANOVA was performed with a significant effect [F (10,14) = 100.9511, p = 4.8·10$^{-8}$]. ANOVAs were followed by a post-hocTukey's test. * p < 0.05, ** p < 0.01, *** p < 0.001, ns not significant. (C, F) The same cells in (A, D) were lysed and a western blot was performed using anti-MICA antibody for detection of MICA and anti-GAPDH antibody as a loading control. Representative of two (C) or three (F) independent experiments. Presence or absence of UL147A and US9 in the different HCMV mutants is annotated. Full experimental data and statistics can be found in S1 Data.

infected with BAC2, BAC20 or BAC20 ΔUS9. Cells were untreated, or incubated for 12h with the translation inhibitor CHX, or with CHX and the proteasomal inhibitor EPX. Cells were lysed at 72 hpi and an immunoblot was performed (Fig 6A). As expected, CHX treatment reduced the quantity of ER-resident MICA\*008 in all cell types, but this form remained detectable in HCMV-infected cells even after 12h of chase, consistent with prolonged retention and/or slow degradation kinetics. Importantly, addition of EPX robustly induced the accumulation of a 37 kDA cytosolic degradation intermediate (annotated as ER-Gly$^{-}$) in cells infected with BAC2 and BAC20, but not BAC20 ΔUS9. These results suggest that US9 and UL147A induce proteasomal MICA\*008 degradation during HCMV infection.

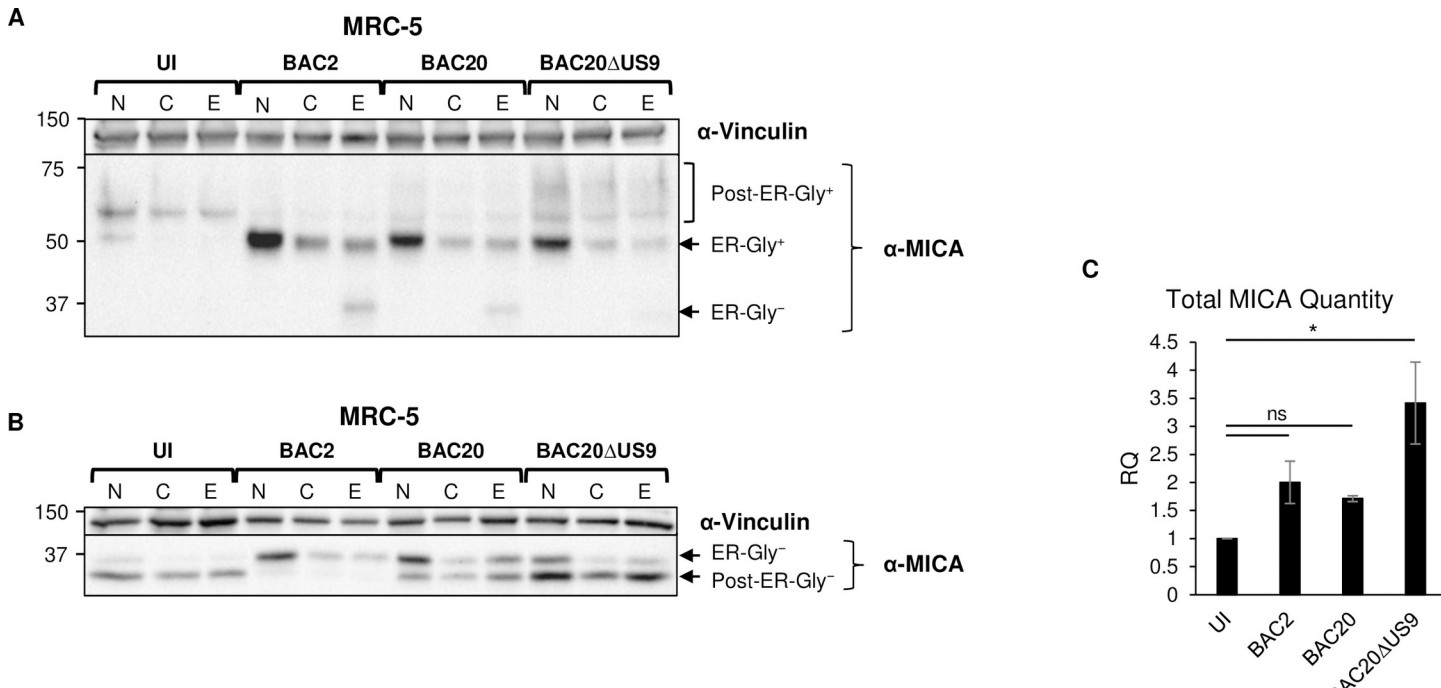

**Fig 6. UL147A and US9 induce maturation arrest but also proteasomal degradation of MICA*008 during HCMV infection.** MRC-5 HLFs (MICA*008 homozygous) were either uninfected (UI) or infected with the indicated HCMV strains. (A-C) Cells were either untreated (N), treated with cycloheximide and DMSO mock treatment (C) or treated with cycloheximide and epoxomicin (E), for 12 hr prior to harvesting at 72 hr post infection (hpi). (A, B) Lysates were prepared and either untreated (A) or deglycosylated with PNGaseF (B). A western blot was performed using anti-MICA antibody for detection of MICA and anti-vinculin antibody as a loading control. (C) Quantification of untreated (N) deglycosylated MICA*008 forms shown in (B). MICA levels were quantified relative to the loading control. RQ, relative quantification. Data show mean ±SEM for three independent experiments. A one-way ANOVA was performed with a significant effect at the p<0.05 level for all conditions [F (3,8) = 6.08, p = 0.018]. A post-hoc Dunnett's test was used to compare UI MICA protein levels to each infected cell. * p < 0.5, ns not significant. Full experimental data and statistics can be found in S1 Data.

To validate the identity of the differently sized MICA*008 forms and facilitate quantitation, we subjected the lysates from the CHX chase assay to digestion with PNGaseF (Fig 6B, untreated cells quantified in Fig 6C). The post-ER, deglycosylated 34-kDa GPI-anchored form of MICA*008 (annotated as post-ER-Gly⁻) was the main form present in UI cells. Following infection with BAC2, this form all but disappeared, and instead the ER-resident, deglycosylated 37-kDa non-GPI-anchored form of MICA*008 (annotated as ER-Gly⁻) became most prevalent. This confirms that MICA*008 maturation is arrested in HCMV infected cells prior to the GPI anchoring phase during infection. The mature, 34 kDa form was partially restored in BAC20-infected cells, and became once more the dominant form of MICA*008 in BAC20 ΔUS9-infected cells, in keeping with measurements of surface MICA*008 expression (Fig 5A and 5B). Nonetheless, considerable amounts of immature, 37 kDa MICA*008 were still present in BAC20 ΔUS9-infected cells compared to UI controls. Notably, there were no differences in deglycosylated mature MICA*008 forms, indicating that the high molecular weight (~70 kDa) MICA*008 forms (Figs 5C and 6A) correspond to mature, GPI-anchored MICA*008 and the increase in size is caused by altered glycan composition in the HCMV-infected cells.

We next quantified total MICA*008 levels in PNGaseF digested, untreated MRC-5 cells (Fig 6C). Overall MICA*008 protein quantity increased significantly by about 3.5-fold in BAC20 ΔUS9-infected cells compared to UI cells, while BAC2 and BAC20 did not differ significantly from UI cells, further supporting degradation of MICA*008 by UL147A and US9. The fold-increase in MICA*008 quantity between BAC2 and BAC20 ΔUS9 was smaller, about

1.7-fold, and did not reach statistical significance. We further assessed total MICA*008 quantity in deglycosylated samples from FLS3*008 cells (S5A and S5B Fig). Similarly, no statistically significant differences were observed between UI and BAC2 cells, but a small yet statistically significant increase of about 1.5-fold in total MICA*008 levels was observed between BAC2 and BAC20 ΔUS9. In summary, the effect of MICA*008 degradation on overall MICA*008 quantity was modest compared to the substantial increase in surface MICA*008 levels in BAC20 ΔUS9-infected cells both in MRC-5 and in FLS3*008 cells (Fig 5B and 5E). We therefore concluded that this increase in MICA*008 surface levels can be mostly attributed to maturation arrest mediated by UL147A and US9.

## UL147A-mediated MICA*008 downregulation leads to reduced NKG2D-mediated killing of HCMV-infected cells

Finally, we asked whether the increase in MICA*008 surface expression in BAC2 ΔUL147A-infected cells would affect NK cell-mediated killing. We performed a killing assay in which MRC-5 HLFs were uninfected or infected with BAC2, BAC2 ΔUL147A or BAC2 ΔUS9. BAC20 and BAC20 ΔUS9 were excluded from this experiment due to the absence of many additional UL*b'*-encoded NK-cell immune evasion functions in these strains, which precludes comparison with the BAC2 single mutants. Infected cells were labelled with radioactive $^{35}$S, harvested at 72 hpi, and co-incubated with primary bulk activated NK cells (Fig 7A). To overcome the variability in killing efficiency, particularly of UI cells, presumably attributable to donor-intrinsic effects, results were normalized to BAC2-infected cells killing levels and a statistical analysis was conducted to compare the effects of infection with the different HCMV strains. MRC-5 cells infected with BAC2 were killed significantly less than BAC2 ΔUL147A or BAC2 ΔUS9, with no significant difference between the two mutants. This indicates that the similar MICA*008 upregulation caused by these deletions (Fig 5A and 5B) was reflected by a comparable increase of NK-cell mediated killing. We then repeated these experiments with NK cells that were either untreated or preincubated with an antibody blocking NKG2D, the activating receptor which binds MICA and other stress-induced ligands (Fig 7B). Importantly, while BAC2 ΔUL147A or BAC2 ΔUS9 both upregulated NK-mediated killing of infected cells by about 1.5-fold, blocking of the NKG2D receptor significantly reduced killing to the point where differences between the mutants and BAC2 were nearly abrogated. This indicates NKG2D is the receptor mediating the observed differential killing. Notably, there were no significant differences between the killing of cells infected with the two mutants, and NKG2D blocking was equally effective for both. We verified that MICA*008 is the only NKG2D ligand differentially expressed between the deletion mutants and BAC2 (S6 Fig), so any NKG2D-mediated differences were attributable to it. Hence, we concluded that the increase in NK-mediated killing of cells infected with BAC2 ΔUL147A or BAC2 ΔUS9 is due to changes in MICA*008 levels, confirming UL147A functionality during HCMV infection.

## Discussion

In this study, we identified UL147A as a MICA*008-specific HCMV immunoevasin, in addition to US9. UL147A is the third MICA-targeting gene discovered in the UL*b'* region after UL142 and UL148A [46], and the fact that three out of six known MICA-targeting genes reside in this region emphasizes its importance for NK cell immune evasion.

Here, we show that UL147A induces MICA*008 maturation arrest and proteasomal degradation. It acts during MICA*008's prolonged and non-canonical maturation process, prior to the GPI-anchoring process (see model in Fig 8). In this respect, UL147A and US9 share

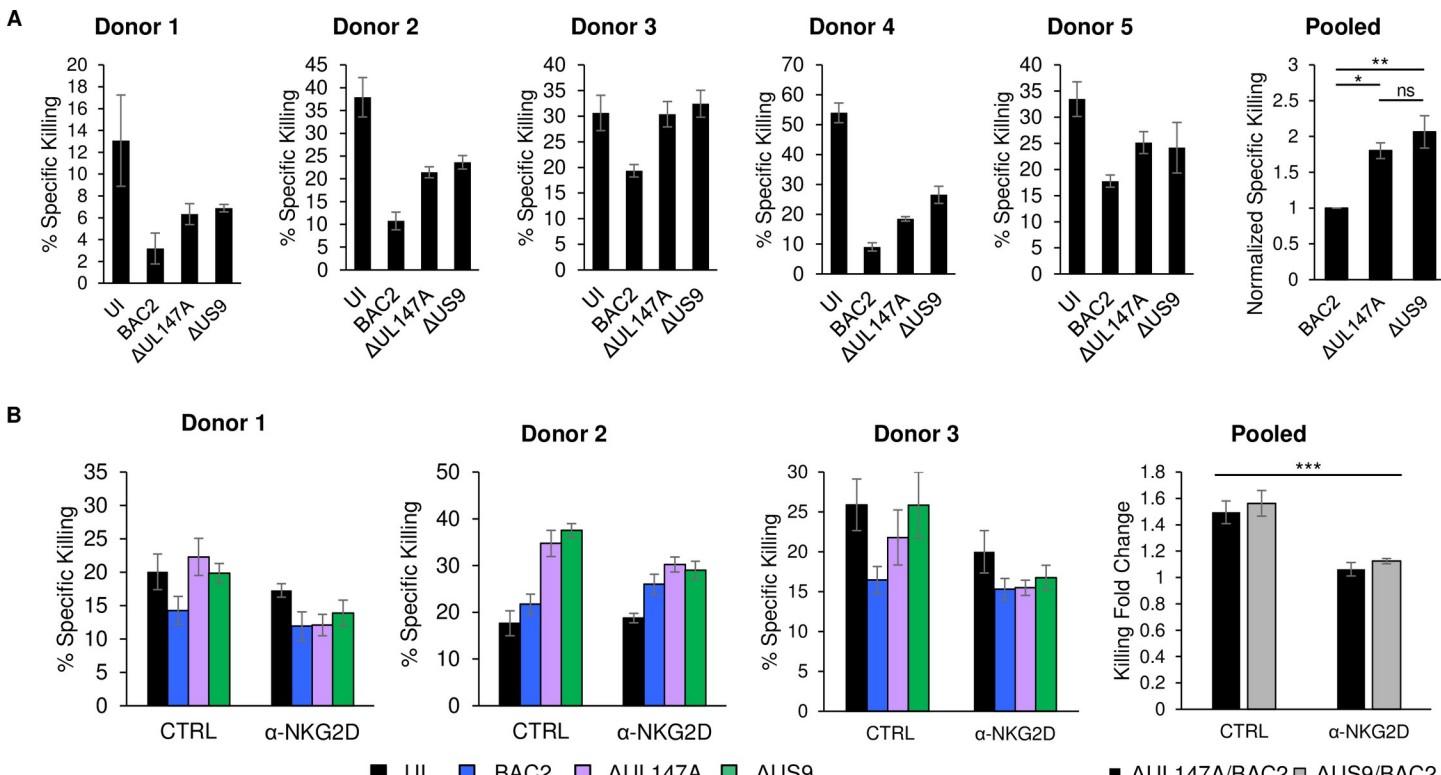

**Fig 7. UL147A-mediated MICA*008 downregulation leads to reduced NK-mediated killing of HCMV-infected cells.** (A) MRC-5 HLFs were uninfected or infected with BAC2, BAC2 ΔUL147A or BAC2 ΔUS9. The cells were radioactively labeled overnight and harvested at 72 hpi, and then co-incubated with NK cells. NK cell-mediated killing was measured by radioactivity release. Shown are five independent experiments from five different NK donors. Data show mean ±STDEV for 3–4 technical replicates for each donor. % Specific Killing values for each donor were normalized to the BAC2 levels and pooled (right panel), data show mean ±SEM. A one-way ANOVA was performed with a significant effect at the p<0.05 level for all conditions [F (2,12) = 10.53, p = 0.0023]. A post-hoc Tukey's test was conducted. * p < 0.05, ** p < 0.01, ns not significant. (B) Killing was performed as in (A), but NK cells were preincubated with an anti-NKG2D antibody to block NKG2D-mediated activation of the NK cells, or left untreated. Shown are three independent experiments from three different NK donors. Data show mean ±STDEV for 4–7 technical replicates for each donor. The fold increase in specific killing of HCMV mutants compared to BAC2 was calculated for each donor and pooled (right panel), data show mean ±SEM. A two-way ANOVA (HCMV mutation, NKG2D blocking) was conducted to compare the change in specific killing. The main effect of NKG2D blocking was significant at the P < 0.05 level [F (1,8) = 38.38, p = 0.00026]. The main effect of HCMV mutation (comparison between ΔUL147A/BAC2 and ΔUS9/BAC2) was not significant [F (1,8) = 0.85, p = 0.38]. There was no significant interaction between NKG2D blocking and HCMV mutation (indicating the mutants were not differentially affected by blocking) [F (1,8) = 0.0017, p = 0.97]. *** p < 0.001. Full experimental data and statistics can be found in S1 Data.

considerable functional homology–both degrade MICA*008 with slow kinetics and require MICA*008 non-canonical GPI-anchoring for recognition.

During HCMV infection, UL147A and US9 act additively, each reducing MICA*008 surface expression by ~2–4 fold, with their combined effect reaching ~10-15-fold reduction. This is reminiscent of US18 and US20 that target full-length MICA, where individual deletions showed modest to no effect on MICA surface expression, but deletion of both dramatically upregulated MICA in a more than additive manner, showing that either protein could partially compensate for the other's absence [36]. US9 and UL147A additivity during infection as opposed to the overexpression model where no such effect was observed, might be due to lower expression levels during infection, so that neither attains its peak effect.

Deletions of UL147A and of US9 significantly increased NKG2D-mediated killing of HCMV-infected cells in a functional assay. We were unable to study the effect of the BAC20 ΔUS9 double deletion mutant on NK cell activation due to its lack of additional UL*b'*-immune evasion encoded functions. One might think that deletion of both genes would have further increased killing, but this would not necessarily be true since NK cells are activated by the

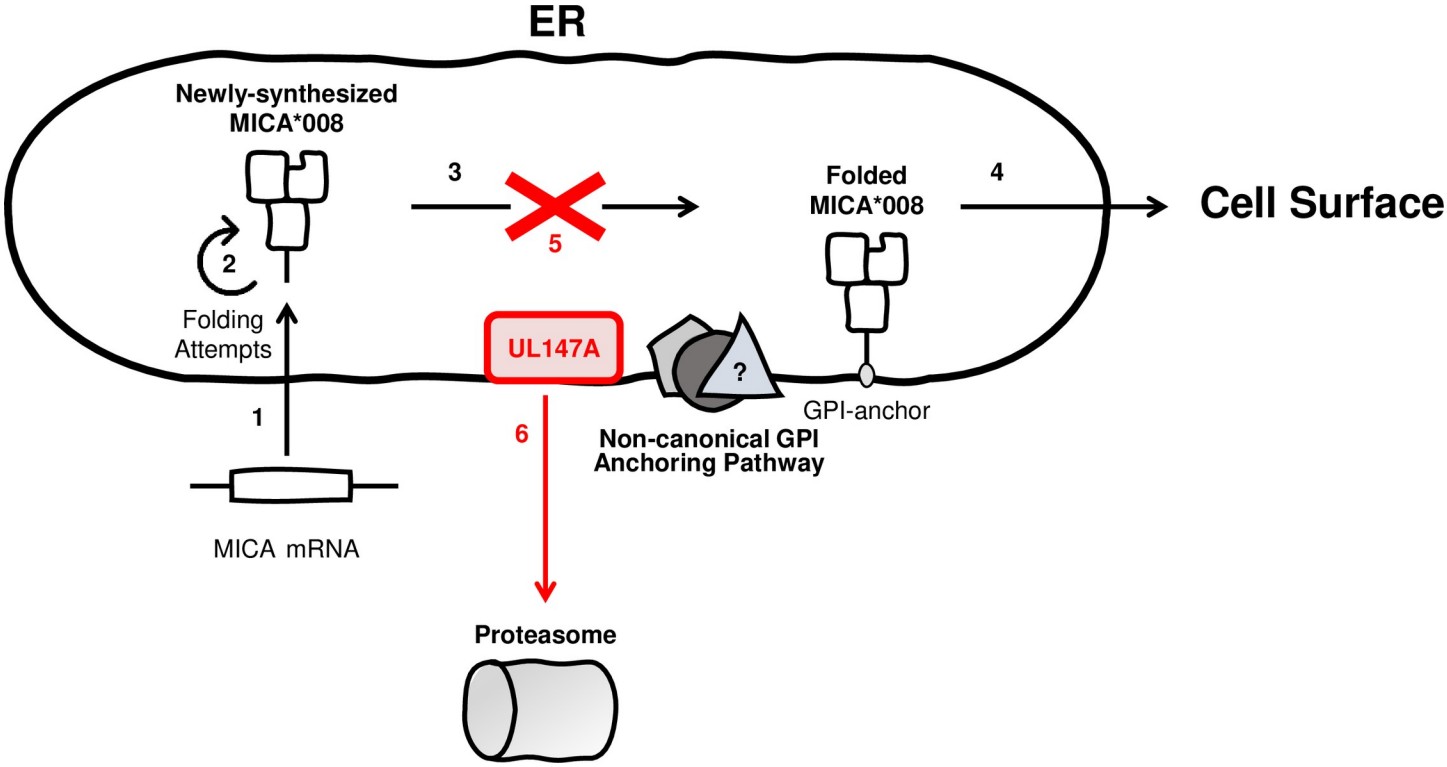

**Fig 8. Model of UL147A Function.** A model of UL147A's effect on MICA*008: (1) following HCMV infection, MICA*008 mRNA is upregulated and the protein is translated into the ER lumen. (2) The immature, non-anchored form of MICA*008 is retained for repeated folding attempts. (3) MICA*008 undergoes GPI anchoring via an unknown non-canonical pathway and (4) subsequently reaches the cell surface. (5) UL147A targets this stage, inhibiting MICA*008 maturation. This is the main effect during HCMV infection. Some non-anchored MICA*008 is subsequently diverted to the cytosol, (6) where it is degraded by the proteasome. Viral mechanisms are marked in red.

integration of diverse and sometimes opposing signals, and do not respond linearly [16]. Indeed, in the case of US18 and US20, deletion of both genes increased NK activation only a little more than the single deletions [36].

In contrast to the robust cell surface effect, individual deletions of UL147A and of US9 failed to significantly change cellular MICA*008 levels, and even a double deletion mutant lacking both proteins only moderately increased total MICA*008 protein levels. Hence, while some MICA*008 is being degraded via the proteasome, the dominant effect of UL147A (and of US9) during infection appears to be maturation arrest in a non-GPI anchored form. However, even in BAC20 ΔUS9-infected cells where MICA*008 was predominantly in the mature form, the immature form still lingered beyond what was observed in UI cells. Whether this signifies the effect of additional MICA*008-sequestering factor(s) or merely reflects changes in MICA*008 production and maturation rates during infection remains unknown.

One possible explanation to the mechanistic discrepancy between overexpression and infection relates to the unusually slow degradation kinetics exhibited by both US9 and UL147A. It may be that in steady-state conditions as in the overexpression system, a new equilibrium is reached with reduced MICA*008 quantities. However, during HCMV infection, MICA expression is strongly induced [36,41,61] perhaps overwhelming this slow process of degradation. Moreover, we recently dissected US9's mechanisms of action [57] and found that it primarily induces MICA*008 degradation indirectly by allowing the cell's physiological ER quality control mechanisms to degrade maturation-arrested MICA*008. These mechanisms are extensively modulated during HCMV infection [62], perhaps impacting their efficiency.

Whether UL147A shares a similar mode of action remains to be addressed. Finally, we cannot rule out the possibility that additional, unknown MICA*008-targeting HCMV immune evasion factor(s) exist and compensate for US9 and UL147A deletions through pathways other than proteasomal degradation.

Despite their functional similarity, UL147A and US9 are encoded in different parts of the HCMV genome and share no significant structural similarity, unlike the neighboring and homologous US18 and US20 [36]. Interestingly, US9 and UL147A are also highly conserved in clinical HCMV strains [42,63]. The only structural feature both proteins have in common is the presence of a short poly-serine stretch in their ER-luminal domains. This implies that the two proteins use different structural elements to induce similar effects. The lack of additive or synergistic interaction between UL147A and US9 in an overexpression system supports the concept that both proteins disrupt similar cellular functions, probably related to the non-canonical MICA*008 maturation pathway. Future studies might utilize the two proteins to uncover this poorly-characterized pathway by seeking either shared or interrelated cellular interaction partners, and thereby also shed light on the mechanistic significance of these structural differences.

Other examples of multiple HCMV immunoevasins targeting a single ligand show complementary mechanisms of action and temporal patterns, aiding in effective suppression of the targeted ligand [25]. However, this is not the case with UL147A and US9, as both are similar in terms of function and temporal activity. While more subtle differences in their mechanism of action may be discovered in the future, another possible explanation for this apparent redundancy is that US9 and UL147A have additional, non-overlapping functions. It was recently shown that US9 also interferes with the STING and MAVS pathways to evade type I interferon responses [64]. A recent proteomic study of the HCMV interactome [4] identified distinct lists of US9 and UL147A interactors. It would therefore be interesting to assess UL147A's ability to regulate other known US9 targets, and to search for additional, unknown targets.

HCMV encodes multiple mechanisms that target MICA: UL142, US18, US20 and UL148A for full-length MICA alleles, and US9 and UL147A for MICA*008 [33,34,36,39,41]. All MICA-targeting mechanisms discovered to date follow a strict dichotomy: full-length allele-targeting or MICA*008-targeting, and UL147A also follows this rule. The division is also maintained with regard to the mechanism of MICA degradation–preferentially lysosomal degradation for full-length alleles versus proteasomal degradation for MICA*008. This division supports the notion that the different mechanisms of degradation might be related to the different biological features of MICA*008 and full-length MICA alleles.

Until recently, MICA*008 was considered an 'escape variant' resistant to MICA-targeting HCMV mechanism [40]. However, we found that US9, and now UL147A, specifically target this prevalent allele. The fact that MICA*008 is targeted by multiple HCMV mechanisms stresses its importance for HCMV biology, and additional MICA*008-targeting mechanisms may be discovered in the future.

MICA itself is only conserved in certain great apes [65], and the truncated mutant MICA*008 is unique to humans [66]. We previously postulated that some of the HCMV mechanisms targeting full-length alleles appeared earlier in CMV evolution, and that following MICA*008's appearance and increasing prevalence, HCMV evolved newer mechanisms to counter this allele [41]. UL147A, which is only conserved in certain great ape CMV, is indeed more recent than US18/20. However, so are full-length-MICA specific UL148A and UL142 [36,43,67]. This indicates continuing evolution of mechanisms that target both full-length MICA alleles and MICA*008.

Another intriguing point to consider is that conserved CMV immune evasion genes may have divergent targets in different species. It was recently shown that the rhesus CMV gene

Rh159 intracellularly retains several simian NKG2D ligands and can also retain human MICA [68]. However, its HCMV homolog, UL148, instead targets CD58 [51,68]. Possibly, UL148 was repurposed as new genes that target the NKG2D ligands arose. It is therefore also possible that MICA*008-targeting genes including UL147A, were also repurposed or gained additional MICA*008-targeting functionality, since their appearance predates that of MICA*008.

In summary, the discovery of UL147A's immune evasion function expands our understanding of the complex and rapidly shifting virus-host interactions which shaped the evolution of the NKG2D ligands.

## Materials and methods

### Cells

The 293T (CRL-3216), HCT116 (CCL-247), RKO (CRL-2577) cell lines were obtained from the ATCC. MRC-5 (CCL-171) primary human lung fibroblasts were also obtained from the ATCC, and VH3 primary human foreskin fibroblasts were obtained from a healthy donor and were previously described [69]. All fibroblasts were used below passage 21. All cell lines and fibroblasts were kept in Dulbecco's modified Eagle's medium, except for MRC-5 cells that were kept in Eagle's minimum essential medium (Biological Industries). Media were supplemented with 10% (vol/vol) fetal calf serum (Sigma-Aldrich) and with 1% (vol/vol) each of penicillin-streptomycin, sodium pyruvate, L-glutamine and nonessential amino acids (Biological Industries). NK cells were isolated from peripheral blood lymphocytes and activated as previously described [70]. NK purity was >95% by FACS analysis. All primary cells were obtained in accordance with the institutional guidelines and permissions for using human tissues.

### Antibodies

The following primary antibodies were used for flow cytometry: anti-MICA (clone 159227; R&D Systems), anti-MICB (clone 236511, R&D Systems), anti-ULBP1 (clone 170818; R&D Systems), anti-ULBP2/5/6 (clone 165903; R&D Systems) and anti-ULBP3 (clone 166514, R&D Systems).

The following primary antibodies were used for immunofluorescence: anti-PDI (ab3672, Abcam), anti-FLAG tag (Clone L5, Biolegend) and anti-MICA (clone 159227, R&D Systems).

The following primary antibodies were used for western blotting: anti-MICA (Clone EPR6568, Abcam), anti-FLAG tag (Clone L5, Biolegend), anti-GAPDH (clone 6C5, Santa Cruz) and anti-vinculin (clone EPR8185, Abcam).

The following secondary antibodies were used: anti-mouse AlexaFluor 647, anti-mouse PE, anti-mouse biotin, anti-rabbit biotin, anti-rat biotin, anti-rabbit Cy3, anti-rat 488, streptavidin-AlexaFluor 647, streptavidin-horseradish peroxidase (HRP), anti-mouse-HRP, anti-rat-HRP and anti-rabbit-HRP, all purchased from Jackson Laboratories.

### Viruses and infection

AD169VarS and AD169VarL were isolated and cloned into BAC2 and BAC20 as previously described [56]. BAC2-generated ULb' block deletion mutants were previously described [49].

Recombinant BAC2 single deletion mutants were generated according to previously published procedures [71,72]. Briefly, a PCR fragment was generated using plasmid pSLFRTKn [73] as the template DNA. The PCR fragments containing a kanamycin resistance gene were inserted into the parental AD169-BAC2 strain [56] by homologous recombination in Escherichia coli. The inserted cassette replaces the target sequence which was defined by flanking sequences in the primers. Recombinant HCMVs were reconstituted from HCMV BAC DNA by Superfect (Qiagen) transfection into permissive MRC-5 cells.

HFFs were used to propagate all HCMV strains and virus stocks were titrated using a plaque assay and stored at -80°C. Infection with the various virus strains was carried out at a multiplicity of infection (MOI) of 2–4, in confluent fibroblasts. HCMV infection was enhanced by centrifugation at 800 g for 30 min.

To verify infection efficiency, the fraction of HCMV-infected cells was assessed by intracellular flow cytometry at 24 hpi. Cells were stained with 0.25 μg/well of an AlexaFluor 488-conjugated anti-CMV-immediate-early antibody (clone 8B1.2; Sigma-Aldrich), and samples were used only if >75% infected and if the variance between samples was <15%.

### Northern blot analysis of specific transcripts

Total RNA was extracted from cells using the RNeasy Mini Kit (QIAGEN). Total RNA was subjected to MOPS gel electrophoresis and transferred to nylon membranes using the Turbo-Blotter (Schleicher and Schuell). Probes were prepared by the DIG High Prime kit (Roche) for detection of indicated transcripts. Hybridization and detection were performed as described by Roche manuals.

### Vectors and primers used for cloning

Generation of the MICA and MICB constructs was previously described [41]. Briefly, sequences were amplified from cDNA of cell lines with the appropriate genotype. Where relevant, site-directed PCR mutagenesis was used to generate MICA and MICB mutants. Sequences were then inserted into lentiviral vector SIN18pRLL-hEFIap-E-GFP-WRPE, replacing the green fluorescent protein (GFP) reporter.

UL147A was cloned from cDNA derived from cells infected with AD169 VarL HCMV strain with the preceding intron sequence, to increase expression levels. To insert a FLAG tag after the endogenous signal peptide, 3 sequential PCR reactions were carried out, with reaction 3 unifying the PCR fragments from reactions 1 and 2 using reaction 1 forward and reaction 2 reverse primers.

The following primers were used for cloning UL147A:

Reaction 1 forward– 5'- CCGCGGCCGCGCCGCCACCTGGAGGCCTAGGCTTTTGC-3' and reaction 1 reverse– 5'-AATCTCCTTGTCGTCATCGTCTTTGTAGTCTGCGAGGATA CTAGTGCTATACCA-3'.

reaction 2 forward– 5'-GACTACAAAGACGATGACGACAAGGAGATTAACGAAAAC TCCTGCTC-3' and reaction 2 reverse– 5'-ggCTCGAGTCAGATCACACAAGTGACGAG GAG-3'

The resultant amplified fragment was cloned into the lentiviral vector pHAGE-DsRED (−)-eGFP(+), which also contains GFP, using the restriction enzymes NotI and XhoI.

### Lentivirus production and transduction

Lentiviral vectors were produced in 293T cells using TransIT-LT1 transfection reagent (Mirus) for a transient three-plasmid transfection protocol as previously described [27]. Cells were transduced in the presence of Polybrene (6 μg/ml). Transduction efficiency was evaluated by GFP levels, and only cell populations with >90% GFP positive cells were used. If necessary, cells were sorted to achieve the required efficiency.

### Flow cytometry

For flow cytometry, cells were plated at equal densities and incubated overnight. Cells were resuspended and incubated on ice with the primary antibody (0.2 μg/well) for 1 h, then

incubated for 30 mins on ice with the secondary antibody (0.75 µg/well). 10,000 live cells were acquired from each sample. In certain experiments, 4′,6-diamidino-2-phenylindole (DAPI) staining was used to exclude dead and dying cells. In all experiments using cells transduced with a GFP-expressing lentivirus, the histograms shown are gated on the GFP-positive population.

## Western blot

Cells were plated at an equal density, incubated overnight, and lysed in buffer containing 0.6% sodium dodecyl sulfate (SDS) and 10-mM Tris (pH 7.4) and the protease inhibitors aprotinin and phenylmethylsulfonyl fluoride (each at 1:100 dilution). In certain cases, lysates were digested with endoglycosidase H (endoH) or peptide N-glycosidase F (PNGaseF, both from NEB), according to the manufacturer's instructions, prior to gel electrophoresis.

Lysates were then subjected to SDS polyacrylamide gel electrophoresis and transferred onto a nitrocellulose membrane. The membrane was blocked in 5% skim milk–phosphate-buffered saline (PBS)–Tween 20 for 1 h and then incubated with a primary antibody overnight, washed 3 times in PBS-Tween 20, incubated with a secondary antibody for 0.5 h, and washed 3 times in PBS-Tween 20. Images were developed using an EZ-ECL kit (Biological Industries). Image Lab software (Bio-Rad) was used for quantification.

## Cycloheximide chase assay and proteasome and lysosome inhibition

This experiment was conducted using RKO cells expressing UL147A fused to an N-terminal 6XHis or FLAG tag or using MRC-5 cells, uninfected or infected with HCMV mutants. For the RKO cells: cells were left untreated or incubated with 50 µg/ml cycloheximide (Sigma-Aldrich) for 8 h, in combination with mock treatment or with the following inhibitors: 100 µg/ml leupeptin (LEU; Merck Millipore), 20 nM concanamycin A (CCMA; Sigma-Aldrich), 8 µM epoxomicin (EPX; A2S), or 8 µM bortezomib (BTZ; LC Biolabs). For the MRC-5 cells: cells were left untreated or incubated with 50 µg/ml cycloheximide (Sigma-Aldrich) for 12 h, in combination with mock treatment or with 2.5µM epoxomicin (EPX; A2S). For all experiments, mock treatment consisted of an equivalent volume of the matching solvent.

## Immunofluorescence

Cells were grown on glass slides, then fixed and permeabilized in cold (−20˚C) methanol. Cells were blocked overnight in CAS-block (Life Technologies) and then incubated overnight with primary antibodies diluted 1:50–200 in CAS block. The next day, cells were washed and incubated overnight in secondary antibodies diluted 1:500 in PBS containing 5% bovine serum albumin. Cells were then washed, treated for 5 min with DAPI, and covered with coverslips. A confocal laser scanning microscope (Zeiss Axiovert 200 M; Carl Zeiss MicroImaging) was used to obtain images, and images were processed using Olympus Fluoview FV1000 software.

## NK cell killing assay

The cytotoxic activity of primary bulk NK cells against HCMV-infected MRC-5 cells was assessed in $^{35}$S release assays as described [70]. NK cells were co-incubated with radioactively-labelled target cells for 5–12 hrs at an E:T ratio of 100:1. In blocking experiments, NK cells were preincubated with RPMI only or with 5 µg per well of anti-NKG2D blocking antibody (clone 149810, R&D Systems) for 1 h prior to co-incubation with target cells. The spontaneous release in all assays was always less than 50% of the total release and is subtracted from the calculation of the percentages of killing. Percentages of killing were calculated as follows: (counts per minute [CPM] sample – CPM spontaneous)/(CPM total – CPM spontaneous) × 100.

## Statistical methods

A one-way ANOVA (with or without repeated measures) was used to compare effects on MICA surface expression (measured as normalized median fluorescent intensity). A Two-way ANOVA was used to compare effects of HCMV infection and NKG2D blocking on killing percentages. The ANOVA was considered statistically significant when $P < 0.05$. Where the ANOVA was statistically significant, post-hoc Tukey's HSD or Dunnett's tests were conducted to determine which mutants differed significantly from each other at alphas of 0.05 or 0.01. Full statistical details including P values, F values, degrees of freedom and effect sizes (Cohen's D) are presented in the relevant figures, figure legends and in S1 Data.

## Supporting information

**S1 Fig. Associated with Fig 1. ΔUL147 deletion mutant does not express UL147A.** (A) MRC-5 cells were infected with the indicated viruses. At 72 hours post infection (hpi), total RNA was isolated. Transcripts were visualized by northern blot analysis using gene-specific probes. The rRNA signals served as a loading control. (B) RKO MICA*008 cells were transduced with an EV, UL147, UL147 N-FLAG, UL147 C-FLAG. MICA surface expression was assayed by flow cytometry. Gray-filled histogram represent a secondary antibody staining of RKO MICA*008 EV cells, similar to all other control stainings. Histogram represents one of 3 biological repeats.
(TIF)

**S2 Fig. Associated with Fig 2. UL147A specifically targets MICA*008.** FACS staining for NK ligands (indicated in the figure) of RKO MICA*008 cells transduced with an empty vector (EV; black histogram) or UL147A (blue histogram). Gray-filled histograms represent secondary antibody staining. Representative of two independent experiments.
(TIF)

**S3 Fig. Associated with Fig 3. UL147A and US9 are redundant in an overexpression model.** (A) RKO MICA*008-HA cells were transduced with an EV, US9-HIS, UL147A-FLAG or with US9-HIS and UL147A-FLAG together to assess synergism between the two. MICA surface expression was assayed by flow cytometry. Gray-filled histograms represent secondary antibody staining of EV cells, all control stainings were similar to the one shown. Representative of three independent experiments. (B) Quantification of MICA surface expression shown in (A), normalized to the EV control. Error bars show mean ±SEM for three independent experiments. A one-way ANOVA was performed to compare the normalized MICA median fluorescence intensity (MFI) between US9, UL147A and the two proteins together. There was no significant effect at the $p < 0.05$ level for all conditions [$F_{(2,6)} = 0.76$, $p = 0.5$]. Full experimental data and statistics can be found in S1 Data.
(TIF)

**S4 Fig. Associated with Fig 5. UL147A kinetics and MICA allele specificity during infection.** (A-B) MRC-5 HLFs (MICA*008 homozygous) were either uninfected (UI) or infected with the indicated HCMV strains. Cells were harvested 24 or 48 hours post infection (hpi). (A) MICA surface expression was assayed by flow cytometry. Gray-filled histograms represent secondary antibody staining of EV cells, all control stainings were similar to the one shown. (B) MICA median fluorescent intensity (MFI) values shown in (A) were quantitated and normalized to BAC2-infected cells. Data show mean ±SEM of three independent experiments. A one way ANOVA was performed. There was no significant effect at the $p < 0.05$ level for all conditions for 24 hpi [$F_{(4,10)} = 1.15$, $p = 0.38$] or for 48 hpi [$F_{(4,10)} = 2.63$, $p = 0.097$]. ns not

significant. (C-D) FLS1 HFFs (endogenous full length MICA*004/*009:01-*049) were either uninfected (UI) or infected with the indicated HCMV strains. Cells were harvested 72 hpi. (C) MICA surface expression was assayed by flow cytometry and the MFI values of three independent experiments were normalized to uninfected cells. A one-way ANOVA was performed with a significant effect at the p<0.05 level for all conditions [F (2,6) = 377.51, p = 4.9·10$^{-7}$], followed by a post-hoc Tukey test. *** p < 0.001, ns not significant. (D) Cells were lysed and a western blot was performed using anti-MICA antibody for detection of MICA, and anti-GAPDH antibody as a loading control. Representative of two independent experiments. Full experimental data and statistics can be found in S1 Data.
(TIF)

**S5 Fig. Associated with Fig 6. UL147A and US9 induce maturation arrest but also reduce MICA*008 quantity during HCMV infection in FLS3*008 cells.** FLS3*008 HFFs (overexpressing MICA*008) were either uninfected (UI) or infected with the indicated HCMV strains. Cells were harvested at 72 hr post infection (hpi) and lysates were prepared (as part of the same experiment shown in Fig 5F). (A) Lysates were deglycosylated with PNGaseF and a western blot was performed using anti-MICA antibody for detection of MICA and anti-vinculin antibody as a loading control. (B) Quantification of deglycosylated MICA*008 forms shown in (A). MICA levels were quantified relative to the loading control. RQ, relative quantification. Data show mean ±SEM for three independent experiments. A one-way ANOVA was performed with a significant effect at the p<0.05 level for all conditions [F (5,12) = 3.51, p = 0.035]. A post-hoc Dunnett's test was used to compare BAC2 MICA protein levels to each infected cell. ** p < 0.01, ns not significant. Full experimental data and statistics can be found in S1 Data.
(TIF)

**S6 Fig. Associated with Fig 7. UL147A-deficient and US9-deficient HCMV mutants are impaired in MICA*008 downregulation only, among NKG2D ligands.** (A-E) MRC-5 HLFs (MICA*008 homozygous) were either uninfected (UI) or infected with the indicated HCMV strains. Cells were harvested 72 hours post infection (hpi). Surface expression of NKG2D ligands was assayed by flow cytometry: MICA (A), MICB (B), ULBP1 (C), ULBP2/5/6 (D), ULBP3 (E). Gray-filled histograms represent a background staining of uninfected cells, similar to all other cells. Representative of two independent experiments.
(TIF)

**S1 Data. Full data and statistics.**
(XLSX)

## Author Contributions

**Conceptualization:** Einat Seidel, Liat Dassa, Dana G. Wolf, Vu Thuy Khanh Le-Trilling, Ofer Mandelboim.

**Formal analysis:** Einat Seidel.

**Funding acquisition:** Ofer Mandelboim.

**Investigation:** Einat Seidel, Liat Dassa, Corinna Schuler, Vu Thuy Khanh Le-Trilling.

**Methodology:** Einat Seidel, Liat Dassa, Esther Oiknine-Djian, Dana G. Wolf, Vu Thuy Khanh Le-Trilling.

**Project administration:** Ofer Mandelboim.

**Resources:** Esther Oiknine-Djian, Dana G. Wolf, Vu Thuy Khanh Le-Trilling.

**Supervision:** Vu Thuy Khanh Le-Trilling, Ofer Mandelboim.

**Visualization:** Einat Seidel, Liat Dassa, Corinna Schuler, Vu Thuy Khanh Le-Trilling.

**Writing – original draft:** Einat Seidel, Liat Dassa, Vu Thuy Khanh Le-Trilling, Ofer Mandelboim.

**Writing – review & editing:** Einat Seidel, Liat Dassa, Vu Thuy Khanh Le-Trilling, Ofer Mandelboim.

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
