## [Decision Letter · Decision Letter 0]

18 Aug 2020

Dear Prof. Mandelboim,

Thank you very much for submitting your manuscript "The human cytomegalovirus protein UL147A downregulates the most prevalent MICA allele: MICA*008, to evade NK cell-mediated killing" for consideration at PLOS Pathogens. As with all papers reviewed by the journal, your manuscript was reviewed by members of the editorial board and by several independent reviewers. In light of the reviews (below this email), we would like to invite the resubmission of a significantly-revised version that takes into account the reviewers' comments.

We cannot make any decision about publication until we have seen the revised manuscript and your response to the reviewers' comments. Your revised manuscript is also likely to be sent to reviewers for further evaluation.

Sincerely,

Richard James Stanton

Guest Editor

PLOS Pathogens

Klaus Früh

Section Editor

PLOS Pathogens

Kasturi Haldar

Editor-in-Chief

PLOS Pathogens

orcid.org/0000-0001-5065-158X

Michael Malim

Editor-in-Chief

PLOS Pathogens

orcid.org/0000-0002-7699-2064

Reviewer's Responses to Questions

**Part I - Summary**

Reviewer #1: Seidel and colleagues describe a role for HCMV UL147A in inhibiting NK cells through the proteasomal degradation of MICA*008, a ligand for NKG2D. To date, a function has not been ascribed to UL147A, while MICA*008 is unusual in that it is a MICA allele with a GPI anchor. UL147A is thus the second HCMV gene that has been shown to target MICA*008. The manuscript also describes mechanistic expts showing that UL147A binds MICA*008 and UL147A function is dependent on the non-canonical system of GPI anchoring specific for MICA*008. The data presented in the main text is convincing as far as it goes, but there are a number of points, addressing of which would give a better physiological context to their data and its interpretation.

Reviewer #2: Seidel et al report another HCMV gene that is able to interfere with the normal expression of MICA*008, an NKG2D ligand which would normally lead to NK cell activation. HCMV is notable for having multiple different viral proteins that interfere with the multiple different human NKG2D ligands and in the case of MICA multiple different alleles. The prevalent MICA allele 008 has some unique features and this group have previously demonstrated that Us9 is able to downregulate it. However, downregulation is incomplete which has led to the current work.

Results are shown with various HCMV strains with a combination of block and single gene deletions the results of which suggest that the ULb’ gene UL147A is also able to specifically target MICA*008. A standard but entirely acceptable set of experiments then show that UL147A is sufficient in an overexpression system, that the protein is degraded, UL147A is ER resident and that UL147A-MICA*008 interact and target MICA*008 for proteasomal degradation. Alterations of MICA*008 domains into MICA*004 show that frameshift and STOP codons are required and transfer of the 008 TM into MICB subsequently allows UL147A targeting.

The requirement for what appears redundant Us9 and UL147A control of MICA*008 is also addressed suggesting that greater levels of MICA*008 are induced in a double negative. A functional NK cell killing assays also show that deletion of either increases killing, however the increased MICA*008 levels with a double KO is not investigated in this functional assay.

The work presented has been well performed and controlled, the data is very clear and well presented, the paper is well written and mostly clear to follow. The targeting NKG2DL for down regulation by a viral gene and targeting to proteosomal degradation is not novel, nor that HCMV has multiple genes that target important immune functions, nor that MICA*008 can be targeted by HCMV. However, the work does convincingly identify another HCMV gene that does this and the mechanism and as such adds to the knowledge about NK cell cell evasion by this virus.

Reviewer #3: The manuscript entitled "The human cytomegalovirus protein UL147A downregulates the most prevalent MICA allele: MICA*008, to evade NK cell-mediated killing" submitted to PLoS Pathogens for publication, explores the contribution of a previously poorly characterized viral gene product on the surface expression of a relatively common MICA allele. Many studies have confirmed that genes non-essential for viral growth are often lost through in vitro passage, yet these serve critical functions in evading the host immune response. UL147A, with is found within the ULb’ region of the HCMV genome, is suggested to play a similar role. It is hypothesized that evasion of NK cell mediate killing that naturally occurs following recognition of MICA*008 is avoided if the virus can prevent its surface expression. The authors of this manuscript have performed a series of elegant biochemical experiments to fully characterize the extent and specificity and mechanism of MICA*008 downregulation by UL147A. The work overall is of high significance to the field of HCMV biology yet the impact to NK biology is less well-developed. However, the data support the authors’ claims, although the overall message of NK cell immune evasion may not be the most accurate description of this work.

This manuscript initially demonstrates how viral mutants downregulate MICA*008 to varying degrees, followed by biochemical studies to quantify and characterize the apparent mechanism. Furthermore, microscopy qualitatively confirms altered distribution of MICA*008 that is due to UL147A expression. Interestingly, the effect of UL147A is only to the MICA*008 allele, and the authors perform a thorough set of experiments, with chimeric constructs, to understand how the non-canonical maturation pathway play a role. Finally, the authors end with an in vitro assay to characterize increased NK cell killing of virus-infected cells deficient in UL147A. However, the reversal of the phenotype of killing is not complete, and the title therefore may be misleading.

Overall, the authors perform the appropriate experiments to assess the biological role of UL147A and it’s effect on MICA*008. A few outstanding questions remain and have been listed below. However, my concern is that both the title and the abstract suggest the that NK cell evasion is a major focus of the manuscript; unfortunately from my perspective it is not. More descriptive information relative to the paper’s findings should be included in the abstract, with slightly less emphasis on NK cell evasion, proportional to the data presented.

**Part II – Major Issues: Key Experiments Required for Acceptance**

Reviewer #1: Broad Points

i. The physiological interpretation of what happens during a HCMV infection cannot be concluded from the data. More detailed points are given below, but broadly, none of the viruses used are wildtype HCMVs, while many of the experiments are performed only using ectopic expression rather than in the context of a HCMV infection.

ii. The NK assays are preliminary.

Detailed points

i. Figure 1 shows a screen of HCMVs and deletion mutants to identify UL147A as targeting MICA*008 and clearly shows this. However there is no inclusion of a virus with a complete UL/b’ region (AD169VarL is missing UL140-144). The authors say they ascertained which gene(s) in the UL/b’ region were responsible, but they haven’t screened the whole region, especially as UL142 also is known to target MICA alleles. This is perhaps of greater significance as the shift caused by AD169VarS looks larger than the shifts by the BAC2 (AD169VarL) mutants.

ii. Figure 1D, F – the shifts are small so an idea of variation through replicates and a statistical test would be stronger rather than a bar chart repeating the flow histogram data.

iii. Figure 2 – This is solid data of the impact of UL147A when ectopically expressed at high level, but what happens during a HCMV infection? Experiments comparing ∆UL147A with parent HCMV should be provided.

iv. Figure 3 – again these experiments are very clean but do not provide information on events during a HCMV infection. UL147A should be tagged in the virus and data provided during a HCMV infection.

v. Figure S2 – Data showing a direct interaction between UL147A and MICA*008 is important and should be provided in the main text if the Figure can be reproduced in a convincing manner (perhaps in the presence of a proteasome inhibitor?). This should also be performed in the context of a HCMV infection.

vi. Figure 5 – The data does not link the effect back to the targeted gene and HCMV genes are multifunctional. An expt with anti-MICA mAb is important to show that it is specifically the effect of knocking out UL147A (or US9) on MICA*008 that causes the increase in susceptibility to NK cells. It may also provide some explanation for the lack of correlation between killing of mock targets and HCMV-infected ones – mocks have hardly any MICA*008 at 72hrs and are killed even more than the BAC2 mutants. A greater range of donors should also be provided allowing statistical testing.

Reviewer #2: (No Response)

Reviewer #3: What is the effect of deleting UL147A on viral growth in vitro? I would suggest performing a multi-step growth curve to show that viral viability is not affected. Maybe UL147A has an additional role in viral replication and thus the partial effect of NK cell killing seen in Figure 5 is due to low viral titers per cell?

**Part III – Minor Issues: Editorial and Data Presentation Modifications**

Reviewer #1: i. MFI is not defined in the Figure legends (median or standard mean or geometric mean?).

ii. Figure 5 - Much more detail is required in the methods – are these NK lines or ex vivo NK cells isolated from PBMC?

Reviewer #2: Results for Fig 1

While reading the figure, given that this group have already described Us9 to have MICA*008 downregulation function I was surprised hat the virus strains used all express Us9 and did wonder what a mutant with both would look like. Clearly, this was addressed in experiments detailed in Figure 5. For clarity the authors might like to add a statement that explains why the exps are done in a US9+ background and that this will be addressed later. Likewise, the abstract/summary does not mention the Us9/UL147A double mutant.

FigS2 demonstrates co-IP data for MICA*008-UL147A I am unsure why it needs to be a supplement and is not panel D in Figure 3 as it completes that analysis and is not really supplemental.

The data shown I Fig5 addresses Us9 and UL147A redundancy and combined function and clearly in a HCMV infection model removal of both via the BAC20 (VarS AD169 short ULb’) delta Us9 construct shows the highest MICA*008 accumulation. Effect on NK cell killing is also addressed, however a double US9 UL147A mutant is not used, I appreciate that delta Us9 on a BAC20 background also deletes other NK evasion genes and would not be suitable for the analysis, but the double mutant could have been made on the BAC2 background to address this? Would the authors predict a further increase in cytotoxicity or not? Figure 5 utilizes a number of BAC HCMV backgrounds as well as additional gene KO the authors should consider adding a little more information to the key to aid the reader – BAC2 is VarL full length Ulb’ while BAC20 is VarS and is deleted for UL147A (as well as most of Ulb’).

Reviewer #3: 1. Since NK cell evasion is a major focus of the paper, why is there no mention in the discussion section of how these findings compare and contrast with similar works?

2. In Figure 1, were VH3 HFFs infected with UL147A deficient HCMV? How does down-regulation compare to VarS?

3. In Figure 1, there are no error bars on the bar plots (D,F). Were these experiments performed only once? There is no mention of replicates in the legend or text.

4. Similarly for Figure 2, S2, and S3, there is no mention of replicate analysis.

5. In Figure 5, red colored bars are not universally applied in the first and second panels of (B) as stated in the legend.

PLOS authors have the option to publish the peer review history of their article (what does this mean?). If published, this will include your full peer review and any attached files.

Reviewer #1: No

Reviewer #2: No

Reviewer #3: No
---

## [Decision Letter · Decision Letter 1]

15 Apr 2021

Dear Prof. Mandelboim,

We are pleased to inform you that your manuscript 'The human cytomegalovirus protein UL147A downregulates the most prevalent MICA allele: MICA*008, to evade NK cell-mediated killing' has been provisionally accepted for publication in PLOS Pathogens.

Best regards,

Richard James Stanton

Guest Editor

PLOS Pathogens

Klaus Früh

Section Editor

PLOS Pathogens

Kasturi Haldar

Editor-in-Chief

PLOS Pathogens

orcid.org/0000-0001-5065-158X

Michael Malim

Editor-in-Chief

PLOS Pathogens

orcid.org/0000-0002-7699-2064

Reviewer Comments (if any, and for reference):

Reviewer's Responses to Questions

**Part I - Summary**

Reviewer #1: The authors have answered all comments in a reasonable fashion. I have no further comments.

Reviewer #2: This is a revised manuscript the authors have addressed all of the comments that I raised in my original review.

Reviewer #3: The authors have adequately addressed my concerns. This revision is a much stronger and more accurate description of the body of work.

**Part II – Major Issues: Key Experiments Required for Acceptance**

Reviewer #1: (No Response)

Reviewer #2: none

Reviewer #3: (No Response)

**Part III – Minor Issues: Editorial and Data Presentation Modifications**

Reviewer #1: (No Response)

Reviewer #2: none

Reviewer #3: (No Response)

PLOS authors have the option to publish the peer review history of their article (what does this mean?). If published, this will include your full peer review and any attached files.

Reviewer #1: No

Reviewer #2: No

Reviewer #3: No

---

## [Editor Report · Acceptance letter]

28 Apr 2021

Dear Prof. Mandelboim,

We are delighted to inform you that your manuscript, "The human cytomegalovirus protein UL147A downregulates the most prevalent MICA allele: MICA*008, to evade NK cell-mediated killing," has been formally accepted for publication in PLOS Pathogens.

Best regards,

Kasturi Haldar

Editor-in-Chief

PLOS Pathogens

orcid.org/0000-0001-5065-158X

Michael Malim

Editor-in-Chief

PLOS Pathogens

orcid.org/0000-0002-7699-2064